# Rotation Invariant Quantization for Model Compression

## Abstract

Post-training Neural Network (NN) model compression is an attractive approach for deploying large, memory-consuming models on devices with limited memory resources. In this study, we investigate the rate-distortion tradeoff for NN model compression. First, we suggest a Rotation-Invariant Quantization (RIQ) technique that utilizes a single parameter to quantize the entire NN model, yielding a different rate at each layer, i.e., mixed-precision quantization. Then, we prove that our rotation-invariant approach is optimal in terms of compression. We rigorously evaluate RIQ and demonstrate its capabilities on various models and tasks. For example, RIQ facilitates $\times 19.4$ and $\times 52.9$ compression ratios on pre-trained VGG dense and pruned models, respectively, with $< 0.4\%$ accuracy degradation. The code is available in the supplementary material.

## 1 Introduction

Deep neural networks are widely used for various tasks, such as computer vision, *Natural Language Processing* (NLP), and recommendation systems. Nevertheless, while performance continuously improves, the models become larger with a massive increase in the number of parameters. In fact, modern *Neural Network* (NN) models may have trillions of parameters, which makes the deployment of these models a challenging task (Chang et al., 2020). One way to mitigate this issue is compressing the model's parameters to reduce its overall memory footprint while satisfying an accuracy constraint. Namely, obtaining a smaller model that is (almost) as capable as the original model.

The most common model compression techniques are weight pruning, quantization, knowledge distillation, and low-rank decomposition. Such optimizations strive to find a smaller model while keeping the original model's accuracy, overlooking the potential inherent in its entropy limit. In the context of NN models, the entropy value is of particular interest as it provides the (theoretical) number of bits required for representing the model parameters. The optimal compression asymptotically attains this entropy limit.

In this context, lossy compression gives considerable merit as it facilitates reducing the NN size significantly with negligible accuracy degradation. The key steps of this approach are the *quantization* and the *encoding* (Polyanskiy & Wu, 2022, Ch. 24). In the quantization phase, the number of unique weight values (symbols) is reduced, consequently reducing the model's entropy. Modifying the symbols' statistics, however, introduces distortion (i.e., quantization error) with respect to the original model. Hence, proper quantization methods are substantial as they determine both the resulting entropy and the distortion of the model's output. In the encoding phase, redundant information is removed, reaching the most compact representation possible without introducing further errors. Finding a solution that quantizes the model at the lowest possible bit rate while satisfying a certain distortion requirement is at the heart of quantization optimization problems, and is known as the rate-distortion problem (Cover & Thomas, 2006).

Typically, NN model quantization optimizes a certain distortion (or accuracy) for a given target rate, which is suitable for reducing the computational complexity of the NN model to a lower precision. However, it is reasonable to consider the dual problem, which optimizes the compression rate while satisfying a certain distortion (or accuracy) requirement. The latter has attained very limited attention in the literature, yet, holds many fascinating challenges in its design, which motivated this study to understand the compression limits in the context of NN.

In this study, we investigate the rate distortion for NN models compression, where the distortion is measured by a cosine distance between the outputs of the original and the quantized model (i.e., angular deviation). In particular, we formulate the model compression as an optimization problem, where the goal is maximizing the model compression ratio subject to a given deviation constraint. Our focus is *mixed-precision solutions*, where each layer gets quantized at a possibly different rate. The advantages of mixed-precision models can be manifested in NLP models, which typically have high redundancy and require extensive memory and loading times. Specifically, the main contribution is as follows.

- We design a post-training Rotation-Invariant Quantization (RIQ) which compresses NN models subject to a deviation constraint. The main theme of our approach is picking the quantization bin width to be proportional to the layers' norm. Since both norm and cosine distance are invariant to rotations, this yields the optimal solution in terms of rate distortion. To find the optimal solution efficiently, we first derive the scale in which the rate increases with the deviation and then suggest a searching paradigm that leverages our finding.

- To analyze the optimality of the RIQ algorithm, we introduce a universal surrogate model that depicts quantization in terms of rotation of the model weights. Its analysis reveals that the rate-distortion minimizing distribution for NN models is a spherical (rotation invariant) distribution constructed by the product of layers' spherical distribution. Due to convexity, the rate achieved under this product distribution is bounded by a rate achieved under the layers' average spherical distribution. Since RIQ optimizes the rate only over spherical distributions, it reaches the optimal solution efficiently.

- We rigorously evaluate the RIQ and demonstrate its capabilities on various models and tasks. RIQ attains a remarkable compression ratio with a negligible accuracy loss in all assessments, surpassing recent results in the literature.

## 2 RELATED WORK

This section is devoted to prior work on model compression that is most relevant to this study. Roughly speaking, typical model compression methods can be classified into four categories. Weight pruning, quantization, knowledge distillation (Sarfraz et al., 2021; Walawalkar et al., 2020), and low-rank decomposition (Idelbayev & Carreira-Perpinán, 2020; Lin et al., 2018; Lee et al., 2019). Even though such methods strive to find a smaller model while retaining the model's accuracy, they often tend to neglect the potential inherent in the entropy limit. In this study, we seek to minimize the model entropy by quantization and then attain this entropy limit by compression while satisfying a distortion requirement at the model's output.

Quantization is a prominent method for compressing NN models. In Wu et al. (2020); Banner et al. (2019); Idelbayev et al. (2021), the authors considered fixed-bit quantization methods, where all layers are quantized at the same integer bit rate. In this paper, on the other hand, we consider post-training mixed-precision solutions. Bhalgat et al. (2020); Wang et al. (2019) focused on quantization-aware training, where the weights quantization is performed during the training. To attain lower quantization rates, Fan et al. (2020); Baskin et al. (2021); Défossez et al. (2022) suggested training the models with noise. Although quantization-aware training methods may achieve better results than the post-training approaches, they are time-consuming, and requires ample representative data and training resources. Nevertheless, these works optimize the distortion, while RIQ optimizes the rate for a given distortion requirement.

The idea of pruning NN connections based on information-theoretic ideas was explored already in the seminal works (LeCun et al., 1989; Hassibi et al., 1993). Later, Han et al. (2015) used magnitude threshold to remove redundant parameters and then utilized Huffman's entropy coding to compress these parameters. Since then, pruning techniques gained popularity, searching for effective methods to prune parameters (Zhang et al., 2021; Frankle & Carbin, 2018; Lee et al., 2019), showing that entropy reduction during training is beneficial, as low-entropy models are more amenable to compression (Oktay et al., 2019; Baskin et al., 2019). Yet, the first rate-distortion formulation of model compression was suggested by Gao et al. (2019); Isik et al. (2022), providing significant insights about the optimal model pruning. In particular, Gao et al. (2019) studied the two-layer networks, providing tractable analysis when assuming the Gaussian distribution of the model weights. This assumption, however, may not hold in general. Isik et al. (2022) observed empirically that

the weights of a pretrained model tend to follow the Laplacian distribution, for which the optimal compression algorithm that resides on the rate-distortion curve must output a sparse model. This study analyzes the rate-distortion of rotation-invariant quantization solutions, which generalizes these findings, as the Gaussian and Laplacian distributions are rotation-invariant distributions.

## 3 PRELIMINARIES

In this section, we rigorously define the model compression optimization problem and the relevant known results on quantization and the rate-distortion theory. Throughout, bold $\mathbf{w}$ denote weight vectors, unless stated otherwise. $\|\cdot\|$ and $\langle\cdot,\cdot\rangle$ denotes the standard $\ell^2$-norm and the inner product, respectively. We use $p_{\mathrm{w}}(w)$ to denote the probability distribution of a random variable w. Hereafter, $\mathbf{w}_{[1:L]} = \{\mathbf{w}_1, ..., \mathbf{w}_L\} \in \mathbb{R}^N$, denotes the concatenation of the weights of a pretrained model with $L$ layers, where $\mathbf{w}_\ell \in \mathbb{R}^{n_\ell}$ are $n_\ell$ weights of layer $\ell$ and $N = \sum_{\ell=1}^{L} n_\ell$. The quantized representation of those weights is denoted by $\hat{\mathbf{w}}_\ell$.

### 3.1 PROBLEM STATEMENT

Let $f : \mathbb{R}^{n_x} \rightarrow \mathbb{R}^{n_y}$ be a *pretrained model* that characterizes the prediction of input $\mathbf{x}$ to an output $\mathbf{y} \in \mathbb{R}^{n_y}$. The model comprises $L$ intermediate mappings called layers, such that $f(\mathbf{x}) = f_{\mathbf{w}_L} \left( f_{\mathbf{w}_{L-1}} (\cdots f_{\mathbf{w}_1}(\mathbf{x})) \right)$, where $\mathbf{w}_\ell \in \mathbb{R}^{n_\ell}$ are the weights of layer $\ell$. We further assume that each layer performs an affine operation (e.g., convolution) followed by a nonlinear operation (e.g., ReLU).

Our goal is to obtain the smallest (quantized and compressed) version of this model $\hat{f}$, whose output is as close as possible to the output of $f$. To assess the fidelity of the quantization, a sample $\mathbf{x}$ is sent through $f$ and $\hat{f}$, and the *deviation between the outputs* $f(\mathbf{x})$ and $\hat{f}(\mathbf{x})$ is measured. In this study, we focus on the cosine distance as distortion measure. That is,

$$d_{f,\mathbf{x}}(\mathbf{w}_{[1:L]}, \hat{\mathbf{w}}_{[1:L]}) \triangleq 1 - \frac{\langle f(\mathbf{x}), \hat{f}(\mathbf{x})\rangle}{\|f(\mathbf{x})\| \cdot \|\hat{f}(\mathbf{x})\|} \tag{1}$$

This distortion reflects the rotation angle that is required to align $\hat{f}(\mathbf{x})$ with $f(\mathbf{x})$. Noticeably, eq. (1) is a rotation-invariant measure, which means that its value does not change when an arbitrary rotation is applied to its arguments. That is, the cosine value does not change when rotating together $\hat{f}(\mathbf{x})$ and $f(\mathbf{x})$. Other distortion measures, such as the $l_2$ and $l_1$ distances, can also be considered for analyzing model compression (Gao et al., 2019; Isik et al., 2022). However, due to the curse of dimensionality, they are mainly useful when addressing low-dimensional optimization (Aggarwal et al., 2001). On the other hand, cosine distance is beneficial to measure the similarity between vectors in high-dimensional spaces, which naturally occurs in models. Further, the correctness of the models in many tasks is determined by the orientation of the output vector rather than the magnitude of the output vector. This motivates us to optimize the quantization rate under the cosine distance measure. Hereafter, the term *deviation* is used to depict $d_{f,\mathbf{x}}(\mathbf{w}_{[1:L]}, \hat{\mathbf{w}}_{[1:L]})$, which is the cosine distance between the outputs, whereas the term *distortion* refers to the cosine distance $d(\mathbf{w}_\ell, \hat{\mathbf{w}}_\ell)$ between the weights $\mathbf{w}_\ell$ and their quantized representation $\hat{\mathbf{w}}_\ell$.

As $\mathbf{x}$ passes the first layer of $f$ and $\hat{f}$, it is rotated (and scaled) by $\mathbf{w}_1$ and $\hat{\mathbf{w}}_1$, respectively. Due to the quantization, $\mathbf{w}_1$ and $\hat{\mathbf{w}}_1$ acts differently on $\mathbf{x}$, and thus, yields unequal outputs. These unequal outputs are rotated (and scaled) by the next layer's weights, and so on, reaching the output of $f$ and $\hat{f}$. The resulting deviation in eq. (1) relates to the distortions gathered through the layers. In particular, each quantized layer produces a rotation distortion in its output, and this distortion keeps propagating and accumulating through the layers until reaching the model's output.

Thus, characterizing the connection between the deviation to the distortion in each layer is a key to optimizing the model quantization. Though finding the exact link is intricate in general, intuitively, as the layers' quantization rate jointly decrease, the deviation increases monotonically with the layers' distortion. Since these distortions are invariant to rotations, the rate-distortion theory tells that the optimal quantization must be rotation invariant as well, as we show in the sequel. This motivates a searching paradigm over rotation-invariant solutions, where the layers' rate are jointly adjusted.

Formally, given a trained model $f$, and a sample $\mathbf{x}$, we wish to find a quantized model $\hat{f}$ whose weights $\hat{\mathbf{w}}_{[1:L]}$ solves the following optimization problem.

$$\min \quad \text{Rate}(\hat{\mathbf{w}}_{[1:L]})$$
$$\text{s.t.} \quad d_{f,x}(\mathbf{w}_{[1:L]}, \hat{\mathbf{w}}_{[1:L]}) \leq D$$

for some deviation requirement $D$, where the rate is the average bits per symbol over the entire quantized model $\hat{f}$. In this work, we characterize the properties of the minimum rate and devise a searching method that finds the minimum rate that satisfies $D$ efficiently.

In this formulation, we consider *mixed-precision quantization* solutions, where the weights $\mathbf{w}_\ell$ of each layer $\ell$ are quantized at a different rate $R_\ell$. In other words, each layer utilizes a different number of symbols. Typically, using fewer symbols induces lower entropy, and hence, we strive to minimize the number of symbols in each layer. Though the entropy indicates the shortest representation of a layer, one must encode the layers' weights to reach this entropy limit. In this study, we utilize *Asymmetric Numeral Systems* (ANS) arithmetic encoder for this purpose (Duda, 2013). Accordingly, the resulting compression ratio, assuming 32 bits representation of the source symbols, is

$$\text{Weights Compression Ratio} \approx \frac{32 \cdot \sum_{\ell=1}^{L} n_\ell}{\sum_{\ell=1}^{L} n_\ell \cdot H(\hat{\mathbf{w}}_\ell) + |T_\ell|} \tag{2}$$

Where $|T_\ell|$ denotes the coding table size of layer $\ell$ and $H(\hat{\mathbf{w}}_\ell)$ is the empirical entropy.

## 3.2 RATE-DISTORTION THEORY

The rate-distortion theory determines the *minimum number of bits per symbol*, or simply the minimum bit rate, required for describing a random variable with a certain (average) distortion. In particular, to quantize a sequence of $n$ independent realizations $\boldsymbol{w} = (w_1, \cdots, w_n)$, generated by a source $\mathbf{w}$ with distribution $p_{\mathbf{w}}(\boldsymbol{w}), \boldsymbol{w} \in \mathcal{W}^n$ into $R$ bits, encoding and decoding functions are utilized. The encoder $\mathcal{E} : \mathcal{W}^n \to \{0, 1\}^R$ maps the sequence to one of $2^R$ possible indices, and the decoder $\mathcal{D} : \{0, 1\}^R \to \hat{\mathcal{W}}^n$ maps the given index into an estimated (quantized) sequence $\hat{\boldsymbol{w}} = (\hat{w}_1, \cdots, \hat{w}_n)$. Thus, the rate-distortion pair $(R, D)$ are the resulting rate $R$ and distance $D = d(\boldsymbol{w}, \hat{\boldsymbol{w}})$ between the original sequence and the quantized sequence.

In general, we wish to minimize both the rate and the distortion, however, there is an inherent tradeoff, characterized by rate-distortion function as (Cover & Thomas, 2006, Ch. 10)

$$R(D) = \min_{p(\hat{\boldsymbol{w}}|\boldsymbol{w}):\mathbb{E}[d(\boldsymbol{w}, \hat{\boldsymbol{w}}))] \leq D} I(\mathbf{w}; \hat{\mathbf{w}}) \tag{3}$$

where $I(\mathbf{w}; \hat{\mathbf{w}}) = H(\mathbf{w}) - H(\mathbf{w}|\hat{\mathbf{w}})$ is the mutual information between the source vector $\mathbf{w}$ and its reconstruction $\hat{\mathbf{w}}$ (Cover & Thomas, 2006, Ch. 2.4), and $d(\cdot, \cdot)$ is a predefined distortion metric, such as the cosine distance in eq. (1). Thus, the rate-distortion function determines the infimum rate $R$ that achieves a given distortion $D$. This infimum is attained by minimizing overall conditional distributions $p(\hat{\boldsymbol{w}}|\boldsymbol{w})$ for which distortion $D$ is satisfied under $p(\boldsymbol{w})$.

## 3.3 UNIFORM SCALAR QUANTIZATION

The rate-distortion theory tells that it is optimal to describe the whole sequence jointly, using one of $2^R$ indices, even when the variables are i.i.d. Yet, in terms of entropy, Koshelev (1963) showed that uniform scalar quantization is (asymptotically) optimal when one intends to further compress the quantized data losslessly. Since this paper considers the latter approach, this section briefly discusses uniform scalar quantization and its analysis.

For a random variable $\mathrm{w} \in [-A/2, A/2]$, where $A \in \mathbb{R}$, uniform quantization partitions the range $[-A/2, A/2]$ into $N$ bins uniformly, such that each bin has width $\Delta = A/N$. Thus, any realization of $\mathrm{w}$ is encoded (rounded) into *an integer value*, $\lceil \mathrm{w}/\Delta \rfloor$, that corresponds to its bin index. The decoder then reconstructs its value by

$$\hat{\mathrm{w}} = \lceil \mathrm{w}/\Delta \rfloor \cdot \Delta \tag{4}$$

The fidelity of this quantization is measured by a distortion measure, such as the *Mean Squared Error* (MSE) criterion, defined as $D(N) = \mathbb{E}|\mathrm{w} - \hat{\mathrm{w}}|^2$. To analyze, it is more convenient to examine the

quantization in terms of rate $R = \log_2 N$. In high-rate regime (i.e., $R \gg 1$), the *probability density* in each bin is nearly flat, and consequently, the distortion is (Polyanskiy & Wu, 2022, Ch. 24.1)

$$\mathbb{E}|\mathbf{w} - \hat{\mathbf{w}}|^2 = \Delta^2/12 \tag{5}$$

Further, the resulting entropy of the quantized symbol is (Cover & Thomas, 2006, Theorem 8.3.1)

$$H(\hat{\mathbf{w}}) = H(\mathbf{w}) - \log(\Delta) \quad \text{[bits/symbol]} \tag{6}$$

where $H(\mathbf{w}) = -\sum_{w \in \mathcal{W}} p(w) \log p(w)$ is the entropy function (Cover & Thomas, 2006, eq. (2.1)). In other words, quantization reduces the entropy by $\log \Delta$, and thus, a larger $\Delta$ yields a lower entropy, and hence, potentially, a higher compression ratio by eq. (2).

## 4 ROTATION-INVARIANT MIXED-PRECISION QUANTIZATION

In this section, we present the RIQ method, which yields a different quantization rate in each layer (i.e., mixed-precision solution) while satisfying the deviation requirement in eq. (1). Then, we use the rate-distortion theory to analyze its performance.

### 4.1 THE RIQ ALGORITHM

Given a model and a deviation requirement, it is sufficient to optimize the rate over rotation-invariant solutions. Typical quantization methods which optimize the distortion for a given rate, on the other hand, determine $\Delta_\ell$ according to the range $\max(\mathbf{w}_\ell) - \min(\mathbf{w}_\ell)$, whose value depends on the orientation of $\mathbf{w}_\ell$, and hence, are not rotation invariant, which is substantial for optimality.

RIQ designs the bin width, $\Delta_\ell$, in proportion to the norm $\|\mathbf{w}_\ell\|$ in each layer. Since norm is invariant to rotations, the resulting $\Delta_\ell$ is indifferent to the orientation of $\mathbf{w}_\ell$. Consequently, the resulting distortion is also indifferent to the orientation of $\mathbf{w}_\ell$, as the bin width dictates the distortion.

Let $\theta_\ell$ be the rotation angle from $\mathbf{w}_\ell$ to $\hat{\mathbf{w}}_\ell$ such that $\langle \frac{\mathbf{w}_\ell}{\|\mathbf{w}_\ell\|}, \frac{\hat{\mathbf{w}}_\ell}{\|\hat{\mathbf{w}}_\ell\|} \rangle \triangleq \cos(\theta_\ell)$. The following lemma examines the relation between $\Delta_\ell$, the norm $\|\mathbf{w}_\ell\|$, and the distortion $d(\mathbf{w}_\ell, \hat{\mathbf{w}}_\ell) = 1 - \cos(\theta_\ell)$.

**Lemma 1.** *Let $\epsilon_\ell \triangleq 1 - \cos(\theta_\ell)$ be the distortion of layer $\ell$. Then, in the high-rate regime, the quantization bin width asymptotically satisfies*

$$\Delta_\ell = \sqrt{\epsilon_\ell} \cdot \|\mathbf{w}_\ell\| \cdot \sqrt{24/n_\ell}$$

The proof of Lemma 1 is elaborated in Appendix A.1. Lemma 1 links between angular distortion, bin width, norm, and dimension. For instance, from the lemma follows that $1 - \cos(\theta_\ell) = \frac{n_\ell \cdot \Delta_\ell^2}{24 \cdot \|\mathbf{w}_\ell\|^2}$, i.e., the distortion scales linearly with the dimension $n_\ell$. Interestingly, this connection expands to the entire model as follows.

**Corollary 1.** *Let $\mathbf{w}_{[1:L]}$ be a vector representation of the weights and $\hat{\mathbf{w}}_{[1:L]}$ denotes its quantized representation, and let $\theta_{[1:L]}$ be the rotation angle from $\mathbf{w}_{[1:L]}$ to $\hat{\mathbf{w}}_{[1:L]}$ such that $\langle \frac{\mathbf{w}_{[1:L]}}{\|\mathbf{w}_{[1:L]}\|}, \frac{\hat{\mathbf{w}}_{[1:L]}}{\|\hat{\mathbf{w}}_{[1:L]}\|} \rangle \triangleq \cos(\theta_{[1:L]})$. Assuming $\|\hat{\mathbf{w}}_\ell\| = \|\mathbf{w}_\ell\| + o(\|\mathbf{w}_\ell\|)$, then,*

$$\cos(\theta_{[1:L]}) = \sum_{\ell=1}^{L} \frac{\|\mathbf{w}_\ell\|^2}{\|\mathbf{w}_{[1:L]}\|^2} \cos(\theta_\ell) + o\left(\frac{\|\mathbf{w}_\ell\|^2}{\|\mathbf{w}_{[1:L]}\|^2}\right)$$

The proof is deferred to Appendix A.2. In words, a rotation of $\mathbf{w}_{[1:L]}$ translates to a convex combination of the layers' rotation, and vice versa. Interestingly, due to convexity of the rate-distortion, it is beneficial to average over as many rotations as possible, which means considering partitioning the model parameters into shorter vectors as we show in the sequel. Still, the most natural partition of the model is simply to it layers, which is considered herein.

The connection of $\Delta_\ell$ to $\|\mathbf{w}_\ell\|$ in Lemma 1 hints at the rotation-invariant nature of the optimization. To focus on rotation-invariant solutions, RIQ introduces a search parameter $k$ that maintains proportion with $\|\mathbf{w}_\ell\|$, allowing efficient search over these solutions. Specifically, when $\Delta_\ell(k) = \|\mathbf{w}_\ell\|/k$ where $k$ to be optimized, the bin-width is indifferent to the orientation of $\mathbf{w}_\ell$ (i.e., rotation invariant). Further, letting the bin width grow linearly with $\|\mathbf{w}_\ell\|$ results in distortion $\epsilon_\ell = 1 - \cos(\theta_\ell) = \frac{n_\ell}{24 \cdot k^2}$, and hence by Corollary 1, $1 - \cos(\theta_{[1:L]}) = \frac{1}{24 \cdot k^2} \sum_{\ell=1}^{L} \frac{n_\ell \cdot \|\mathbf{w}_\ell\|^2}{\|\mathbf{w}_{[1:L]}\|^2}$. Namely, the distortion of the entire parameters scales as $O(1/k^2)$. Remarkably, the deviation also scales as $O(1/k^2)$.

---

**Algorithm 1** The RIQ algorithm

---

**Input:** model weights $\mathbf{w}_{[1:L]}$, deviation requirement $D$, minimum error $\epsilon_0$

Initialize $k_{\min} = \frac{\sqrt{n_{\ell^*}/24}}{1-\epsilon_0}$, $\qquad k_{\max} = \frac{\sqrt{n_{\ell^*}/24}}{\sqrt{\epsilon_0}\cdot\epsilon_0}$, $\qquad k = k_{\min}$, $\qquad \text{step} = \sqrt{k_{\max} - k_{\min}}$.

**while** $k \le k_{\max}$ **do**
    **for** $\ell = 1$ **to** $L$ **do**
        set $\Delta_\ell = \|\mathbf{w}_\ell\| \cdot \left( \frac{1}{k} + \epsilon_0 \cdot \sqrt{\frac{24}{n_\ell}} \right)$ and quantize by $\hat{\mathbf{w}}_\ell = \left\lceil \frac{\mathbf{w}_\ell}{\Delta_\ell} \right\rceil \cdot \Delta_\ell$
    **end for**
    **if** $d_{f,x}(\mathbf{w}_{[1:L]}, \hat{\mathbf{w}}_{[1:L]}) \le D$ **then**
        **if** step $\le 3$ **then**
            compress to $H(\hat{\mathbf{w}}_{[1:L]})$ with entropy encoder
        **end if**
        $k_{\max} = k$ ; $\qquad$ step $= \sqrt{\text{step}}$; $\qquad k = k - \text{step} \cdot \lfloor \text{step} \rfloor$
    **end if**
    $k = k + \text{step}$
**end while**

---

**Proposition 1.** *In the high rate regime, the deviation in eq.* (1) *under RIQ scales as* $O(1/k^2)$.

The proof is deferred to Appendix A.3. Essentially, since $\Delta_\ell(k)$ and $\epsilon_\ell$ are monotonically decreasing with $k$, then by eq. (6), the entropy increases with $k$. This allows RIQ is to reach the smallest $k$ (i.e., minimum entropy) solution that satisfies the deviation requirement in eq. (1).

Next, we introduce an efficient iterative searching algorithm for finding the optimal $k$. In each iteration, the algorithm refines the searching range until reaching the smallest $k$ (up to a small constant) that satisfies the deviation requirement. Practically, however, as $k$ increases, $\Delta_\ell(k) \to 0$. To prevent this, we add a small constant $\epsilon_0$ to $\sqrt{\epsilon_\ell}$, which bounds the value of the smallest $\Delta_\ell(k)$. In this case, setting $\sqrt{\epsilon_\ell} = \frac{1}{k}\sqrt{\frac{n_\ell}{24}} + \epsilon_0$, yields,

$$\Delta_\ell(k) = \|\mathbf{w}_\ell\| \cdot \left( \frac{1}{k} + \epsilon_0 \cdot \sqrt{\frac{24}{n_\ell}} \right) \tag{7}$$

Even though the search of $k$ is unbounded in general, practically it is sufficient to search in bounded space since the weights' norm is finite (Idelbayev et al., 2021). In the following proposition, we derive searching bounds for the optimal $k$. Let $k^*$ be the optimal (smallest) $k$ that satisfies constraint $D$, and let $\ell^*$ be the index of the layer with the largest $n_\ell$ in $f$.

**Proposition 2.** *The optimal* $k^*$ *satisfies the following bounds:* $\frac{\sqrt{n_{\ell^*}/24}}{(1-\epsilon_0)} \le k^* \le \frac{\sqrt{n_{\ell^*}/24}}{(\epsilon_0 \cdot \sqrt{\epsilon_0})}$.

The proof is deferred to Appendix A.4. To further improve the search time, a nested refinement is utilized. Specifically, at each stage, only $\sqrt{|O(k)|}$ values of $k$ in ascending order are considered. Once a certain value of $k$ satisfies the deviation requirement $D$, this $k$ becomes the new upper limit for the search, and the search region is refined within a smaller region of $k$ to consider, again with only $\sqrt{|O(k)|}$ potential values to inspect. This repeatedly continues until the search step is sufficiently small (e.g., step $\le 3$), and the compression gain becomes negligible. These refinements enable fast convergence in relatively few iterations. See Algorithm 1 for a detailed description of RIQ.

**Remark.** *The additional degree of freedom that* $\epsilon_0$ *gives is substantial. For example, it facilitates enforcing quantization to a maximum of $R$ bits (e.g., $R = 8$ bits) for low precision runtime, by setting the limit $k \to \infty$ at* $\epsilon_0(\ell) = \frac{\max(\mathbf{w}_\ell) - \min(\mathbf{w}_\ell)}{2^R - 1} / \sqrt{\frac{24 \cdot \|\mathbf{w}_\ell\|^2}{n_\ell}}$.

For simplicity, in the sequel we apply the same small common constant value $\epsilon_0$ to all layers.

### 4.2 RIQ RATE-DISTORTION ANALYSIS

In this section, we provide theoretical justification for the optimality of RIQ. We introduce a surrogate model for which the rate-distortion analysis with cosine distance is tractable, showing that the minimizing distribution of the mutual information is indifferent to the orientation of $\mathbf{w}_\ell$, and is characterized by a single parameter $k$, as RIQ suggests.

First, extending eq. (4) to NN model quantization, where layer $\ell$ is encoded uniformly, yields

$$\hat{\mathbf{w}}_\ell = \lceil \mathbf{w}_\ell / \Delta_\ell \rfloor \cdot \Delta_\ell \tag{8}$$

For tractability, it is common to analyze the rate-distortion for eq. (8) by a surrogate model in which the distortion is modeled as a random additive noise (Kipnis & Reeves, 2021; Marco & Neuhoff, 2005). Yet, when considering angular deviation such representation hinders the rate-distortion analysis since the relation between additive noise to cosine distance that we wish to examine is intricate. Accordingly, we suggest a reparameterization to the additive noise model, which represents quantization as *random rotation* (and scale) of the weights in each layer. This enables to analyze the rate-distortion of eq. (8) for the deviation in eq. (1) in the sequel.

**Surrogate Model.** *Let $\mathbf{w}_\ell$ be the weights of layer $\ell$, and let $\hat{\mathbf{w}}_\ell$ denote their quantized representation. Let $\theta_\ell$ be a random rotation angle from $\mathbf{w}_\ell$ to $\hat{\mathbf{w}}_\ell$, such that $\langle \frac{\mathbf{w}_\ell}{\|\mathbf{w}_\ell\|}, \frac{\hat{\mathbf{w}}_\ell}{\|\hat{\mathbf{w}}_\ell\|} \rangle = \cos(\theta_\ell)$, and let $\mathbf{U}(\theta_\ell | \mathbf{w}_\ell) \in SO(n_\ell)$ be a random orthogonal transformation corresponding to a random rotation that is $\theta_\ell$ away from $\mathbf{w}_\ell$. Then,*

$$\tilde{\mathbf{w}}_\ell = \|\hat{\mathbf{w}}_\ell\| \cdot \mathbf{U}(\theta_\ell | \mathbf{w}_\ell) \frac{\mathbf{w}_\ell}{\|\mathbf{w}_\ell\|} \tag{9}$$

*models the quantized weights $\hat{\mathbf{w}}_\ell$.*

Intuitively, $\mathbf{U}(\theta_\ell | \mathbf{w}_\ell)$ randomly rotates any given vector uniformly on a sphere, where one degree of freedom is lost due to the requirement of being $\theta_\ell$ away from $\mathbf{w}_\ell$. To obtain $\hat{\mathbf{w}}_\ell$ in eq. (8), the realization of $\mathbf{U}(\theta_\ell | \mathbf{w}_\ell)$ should rotate the unit vector $\mathbf{w}_\ell / \|\mathbf{w}_\ell\|$ in the plane generated by $\mathbf{w}_\ell$ and $\hat{\mathbf{w}}_\ell$, and then, stretches it into the length $\|\hat{\mathbf{w}}_\ell\|$ (see Figure 3(a) for illustration).

In other words, this model describes a random vector $\tilde{\mathbf{w}}_\ell$ that is uniformly distributed on a cone that is $\theta_\ell$ away from $\mathbf{w}_\ell$, for which a single realization matches eq. (8). The merit of this model is its tractable analysis, from which *spherically symmetric distribution* emerges to depict the quantized weights (Fang et al., 2018, Definition 2.1). Accordingly, each layer obtains a randomly rotated version of $\mathbf{w}_\ell$, which translates to a joint rotation at angle $\theta_{[1:L]}$ of all the parameters by Corollary 1. Consequently, in the high rate regime, where the support of $\theta_\ell$ is sufficiently small, the distortion and the deviation decrease at same scale with $k$ by Proposition 1.

**Proposition 3.** *Let $\mathbf{w}_\ell$ be the weights of layer $\ell$, and let $\tilde{\mathbf{w}}_\ell$ model the quantized representation of those weights, modeled by eq. (9). Then, $\tilde{\mathbf{w}}_\ell | \mathbf{w}_\ell$ has a spherical (rotation-invariant) distribution.*

A detailed proof is given in Appendix A.5. Essentially, the strength of Proposition 3 is twofold. First, it proves that the distribution of $\tilde{\mathbf{w}}_\ell | \mathbf{w}_\ell$ in each layer does not change when arbitrary rotations are applied to it. Second, it holds for any distribution of $\mathbf{w}_\ell$ and $\theta_\ell$. This universality is substantial when considering various models and tasks. The following theorem extends the results of Proposition 3 to multiple layers, showing that spherical distribution is also the rate-distortion minimizing distribution.

**Theorem 1.** *Let $f$ be a NN model with $L$ statistically independent layers whose weights are $\mathbf{w}_{[1:L]}$, and let $\tilde{\mathbf{w}}_{[1:L]}$ be their quantized representation. Then, the unique minimizing distribution $p(\tilde{\boldsymbol{w}}_{[1:L]} | \boldsymbol{w}_{[1:L]})$ of the rate-distortion function*

$$R(D) = \min_{\substack{p\left(\tilde{\boldsymbol{w}}_{[1:L]} \middle| \boldsymbol{w}_{[1:L]}\right): \\ \mathbb{E}\left[d_{f,x}(\mathbf{w}_{[1:L]}, \tilde{\mathbf{w}}_{[1:L]})\right] \le D}} I\left(\mathbf{w}_{[1:L]}; \tilde{\mathbf{w}}_{[1:L]}\right) \tag{10}$$

*is a product distribution constructed as the product of the layers' spherical distribution. Consequently, the infimum rate is characterized by a single parameter.*

The detailed proof is given in Appendix A.6. The key steps of the proof are, first, showing that the minimizing distribution is a product distribution. Then, due to the convexity of the rate-distortion in eq. (10), we bound the mutual information with a convex combination of $\theta_\ell$ distributions, given in Corollary 1. Consequently, the problem is simplified to a single (average) layer optimization, which is governed by a single quantization parameter.

Remarkably, the joint minimizing distribution of the model's weights $p\left(\tilde{\boldsymbol{w}}_{[1:L]} | \boldsymbol{w}_{[1:L]}\right)$ is also spherical since any partitioning of spherical distribution (naturally occurring by the model's layers) remains spherical (Fang et al., 2018, Theorem 2.6). Further, due to the convexity of the mutual information, it

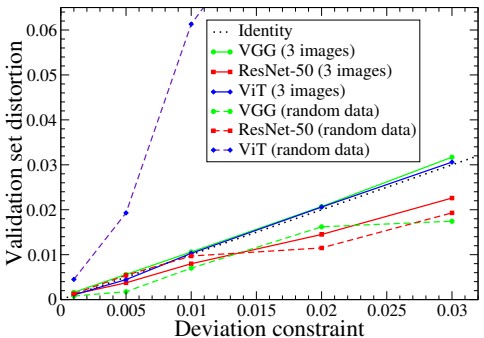 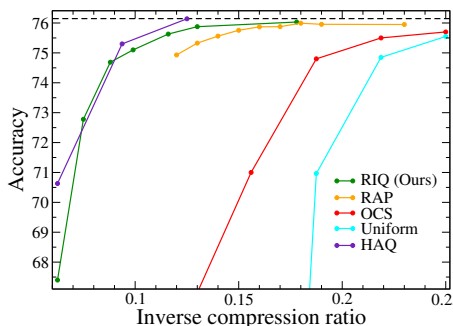

Figure 1: (a) Cosine distance of the validation dataset as a function of the deviation constraint (on the calibration dataset). Models include VGG (green), ResNet-50 (red), and ViT (blue). (b) Accuracy vs Compression for ResNet-50 model. RIQ (green) vs. RAP (orange), OCS (red), linear quantization (cyan), and HAQ (purple). HAQ, however, requires training after the quantization.

is beneficial to consider as many partitions as possible, which can only reduce the mutual information. Practically, however, running the model layer-by-layer, where each layer is quantized by a single scalar, has merit due to its simplicity. We point out that the minimizing distribution cannot be found explicitly without any assumption on the distribution of the weights. Interestingly, the optimal solutions presented by Gao et al. (2019); Isik et al. (2022) for the Gaussian and Laplace distributions coincide with our rotation-invariant observation as these distributions are spherical distributions as well. Characterizing a universal rate-distortion problem is substantial as it reveals that it is sufficient to consider only rotation-invariant solutions for any model and task.

## 5    EMPIRICAL RESULTS

In this section, we evaluate the compression ratio and model accuracy of RIQ with ANS and compare them to relevant baseline results. In all experiments, we use pre-trained models for their relevant tasks. Specifically, for classification tasks we use VGG, Simonyan & Zisserman (2014), ResNet-50, He et al. (2016), and ViT, Dosovitskiy et al. (2020) models, from the torchvision library[1] on the ImageNet data-set (I1k). For detection tasks, we use YOLOv5, Jocher et al. (2022), from Neural-Magic library [2] on the COCO dataset (Lin et al., 2014). For NLP task, we use the DistilBERT model, Sanh et al. (2019), on SQuAD dataset, (Rajpurkar et al., 2016). Following RIQ, an ANS encoder encodes each quantized layer to its entropy limit. An efficient implementation of ANS on GPU was demonstrated by Weißenberger & Schmidt (2019), reaching impressive decoding rates. Consequently, RIQ facilitates optimizing and compressing models in a few minutes. For reproduction purposes, we provide a Python code of our algorithm as a supplementary material which includes both the quantization phase (RIQ) and compression phase (ANS). Additional results are given in Appendix A.8.

To measure the resulting deviation by eq. (1) on a validation set as a function of a deviation requirement $D$, we use two types of calibration data: (a) three real images, sampled from the training, and (b) randomly generated data that follows the Gaussian distribution. In Figure 1(a) we present the deviation measurements on three models: ResNet-50, VGG, and ViT, where the identity line (black-dotted) is given for reference. As we see, even a small calibration set of three images (solid lines) is sufficient to predict the deviation on the validation set. Further, we see that the randomly generated data may not predict well the resulting deviation on the validation set, leading to either a less compressed model (ResNet-50 and VGG) or a higher deviation (ViT).

In Figure 1(b), we evaluate the effect of RIQ on the accuracy and the inverse compression ratio (i.e., the reciprocal of eq. (2)) for a pre-trained ResNet-50 model. Interestingly, the rate-distortion curve reflects well the accuracy-compression tradeoff. For comparison, we depict the accuracy-compression results of the *Relaxed Advanced Pipeline* (RAP) method, Hubara et al. (2021), *Outlier Channel Splitting* (OCS), Zhao et al. (2019), and the *Hardware-aware Automated Quantization* (HAQ), Wang et al. (2019), which requires further training for fine-tuning. Indeed, RIQ surpasses other post-training

---

[1] pytorch.org/vision/stable/models.html    [2] sparsezoo.neuralmagic.com

quantization methods, yet, falls short compared to HAQ. This is since retraining the quantized model yields a different rate-distortion curve, which is out of the scope of this paper.

Table 1: Comparison of Top-1 accuracy on ImageNet for various baselines.

| Model | Compression | Method | Acc. (%) | Ref. (%) | Drop (%) |
|---|---|---|---|---|---|
| VGG-16 | ×10.6 (3 bits) | GPFQ Zhang et al. (2022a) | 70.24 | 71.59 | 1.35 |
| | | **RIQ (Ours)** | **71.58** | **71.59** | **0.01** |
| | ×8 (4 bits) | MSE Banner et al. (2019) | 70.50 | 71.60 | 1.10 |
| | | OMSE Choukroun et al. (2019) | 71.48 | 73.48 | 2.00 |
| | | GPFQ Zhang et al. (2022a) | 70.90 | 71.59 | 0.69 |
| | | **RIQ (Ours)** | **71.55** | **71.59** | **0.04** |
| | ×6.4 (5 bits) | GPFQ Zhang et al. (2022a) | 71.05 | 71.59 | 0.54 |
| | | **RIQ (Ours)** | **71.58** | **71.59** | **0.01** |
| ResNet-50 | ×16 (2 bits) | SuRP with tuning Isik et al. (2022) | **76.4** | **76.6** | **0.2** |
| | | **RIQ (Ours)** | 69.47 | 76.14 | 6.67 |
| | ×10.6 (3 bits) | GPFQ Zhang et al. (2022a) | 70.63 | 76.13 | 5.50 |
| | | **RIQ (Ours)** | **74.76** | **76.14** | **1.38** |
| | ×8 (4 bits) | MSE Banner et al. (2019) | 73.80 | 76.10 | 2.30 |
| | | OMSE Choukroun et al. (2019) | 73.39 | 76.01 | 2.62 |
| | | AdaRound Nagel et al. (2020) | 75.23 | 76.07 | 0.84 |
| | | S-AdaQuant Hubara et al. (2021) | 75.10 | 77.20 | 2.10 |
| | | BRECQ Li et al. (2021) | 76.29 | 77.00 | 0.71 |
| | | GPFQ Zhang et al. (2022a) | 74.35 | 76.13 | 1.78 |
| | | **RIQ (Ours)** | **75.61** | **76.14** | **0.53** |
| | ×6.4 (5 bits) | GPFQ Zhang et al. (2022a) | 75.26 | 76.13 | 0.87 |
| | | **RIQ (Ours)** | **75.95** | **76.14** | **0.19** |

In Table 1, we compare RIQ to relevant baseline methods on the VGG-16 and ResNet-50 models. In this table, we modified RIQ in Algorithm 1 to minimize the deviation (accuracy drop) for a given rate requirement. Noticeably, RIQ outperforms most of the baselines, as it reaches the entropy limit by the ANS. Applying ANS to other baselines, however, can only degrade their compression since they consider per-channel quantization, for which the encoding table overhead becomes significant. Further, applying ANS to their layers still degrades compression as the number of unique symbols per-layer is much larger than per-channel. Moreover, some baselines quantized both weights and activation to further accelerate the models. The contribution of ANS to RIQ is discussed in Appendix A.8.

Typical compression ratio and score achieved by RIQ are presented in Table 2 for a variety of models and tasks. Note that the deviation does not translate immediately to the drop in each score, as the latter is a task-specific measure. Yet, in general, the scores improve as the deviation decreases. To further assess the potential of RIQ, we evaluate our method on sparse models taken from the Neural-Magic[2]. Notably, the compression ratio of sparse models is significantly higher. This coincides with the conclusion that pruning is substantial for good compression (Isik et al., 2022).

Table 2: Compression and accuracy achieved by RIQ. Models denoted by asterisk (*) were pruned during training, before quantization.

| Model / Dataset | Metric | Deviation constraint | Compression | Quant. Score | Ref. Score | Drop |
|---|---|---|---|---|---|---|
| VGG / I1k | Top-1 Acc (%) | 0.5% | ×**19.4** | 71.3 | 71.59 | **0.29** |
| ResNet-50 / I1k | Top-1 Acc (%) | 0.5% | ×**7.31** | 75.88 | 76.14 | **0.26** |
| ViT / I1k | Top-1 Acc (%) | 0.5% | ×**6.98** | 81.0 | 81.07 | **0.07** |
| YOLO / COCO | mAP@.5 | 0.3% | ×**8.34** | 54.7 | 55.7 | **1.0** |
| DistilBERT / SQuAD | F1 | 0.025% | ×**7.96** | 85.0 | 85.08 | **0.08** |
| VGG*(75%) / I1k | Top-1 Acc (%) | 0.5% | ×**52.9** | 69.34 | 69.73 | **0.39** |
| ResNet-50*(95%) / I1k | Top-1 Acc (%) | 0.5% | ×**41.5** | 75.72 | 76.14 | **0.42** |
| YOLO*(75%) / COCO | mAP@.5 | 0.3% | ×**16.48** | 52.6 | 53.5 | **0.9** |
| DistilBERT*(58%) / SQuAD | F1 | 0.025% | ×**19.4** | 84.70 | 84.92 | **0.22** |

## 6 CONCLUSION

In this paper, we have investigated a post-training quantization method that strives to minimize the rate of the model's parameters subject to a deviation constraint. A *rotation-invariant quantization* scheme (RIQ) was introduced, which quantizes each layer in proportion to the layer's norm, searching for the optimal solution over the family of spherical distributions. To find the solution efficiently, we derived the scale in which the rate increases with the deviation and then suggest a searching paradigm that bounds the search space based on our findings. The rate-distortion curve was thoroughly analyzed, showing that the minimizing distribution is a product distribution, constructed as the product of the layer's spherical distribution, which coincides with the RIQ approach.

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

# A APPENDIX

In this section, we provide rigorous proofs for the theorems and the statements herein. Further, we present additional results for RIQ.

## A.1 PROOF OF LEMMA 1

*Proof.* Let $\mathbf{w}_\ell$ be the realization of the weights vector of layer $\ell$, and $\hat{\mathbf{w}}_\ell$ be the quantized representation of those weights, where $\theta_\ell$ denotes the angle between those vectors. Before diving into the cosine distance analysis, let us revisit the mean squared error analysis of the uniform quantizer in Section 3.3, and extend it to the multivariate case. In this case, the distortion is

$$\|\mathbf{w}_\ell - \hat{\mathbf{w}}_\ell\|^2 = n_\ell \cdot \frac{1}{n_\ell} \sum_{i=1}^{n_\ell} |\mathbf{w}_{\ell,i} - \hat{\mathbf{w}}_{\ell,i}|^2$$
$$\stackrel{(a)}{=} n_\ell \cdot \mathbb{E} |\mathbf{w}_{\ell,j} - \hat{\mathbf{w}}_{\ell,j}|^2 + o(1)$$
$$\stackrel{(b)}{=} n_\ell \cdot \Delta_\ell^2/12 + o(1) \tag{11}$$

where (a) follows from the law of large numbers, and (b) follows by the analysis of the scalar uniform quantizer, given in (Polyanskiy & Wu, 2022, Ch. 24.1).

Note that the quantization error in each layer can be bounded by $\|\mathbf{w}_\ell - \hat{\mathbf{w}}_\ell\| \leq \sqrt{n_\ell} \frac{\Delta_\ell}{2}$. Assuming that the range of $\mathbf{w}_\ell$ is symmetric around zero for simplicity, then, $\Delta_\ell \approx \frac{2\max(\mathbf{w}_\ell)}{2^R}$, where $R$ is the rate of quantization. Hence, the error can be bounded by $\|\mathbf{w}_\ell - \hat{\mathbf{w}}_\ell\| \leq \sqrt{n_\ell} \frac{\max(\mathbf{w}_\ell)}{2^R} \approx \frac{\|\mathbf{w}_\ell\|}{2^R}$, which is $O(\|\mathbf{w}_\ell\|)$. Yet, this $O(\|w\|)$ does not reflect well the high-rate regime that this paper considers. In particular, to focus on the high-rate regime, we let $R = O(\log^c(\|\mathbf{w}_\ell\|))$ for $c > 1$, which results in error bound of $o(\|\mathbf{w}_\ell\|)$.

For analyzing the cosine distance between $\mathbf{w}_\ell$ and $\hat{\mathbf{w}}_\ell$ in the high-rate regime, we notice that

$$\|\mathbf{w}_\ell - \hat{\mathbf{w}}_\ell\|^2 = \|\mathbf{w}_\ell\|^2 + \|\hat{\mathbf{w}}_\ell\|^2 - 2\|\mathbf{w}_\ell\| \cdot \|\hat{\mathbf{w}}_\ell\| \cos(\theta_\ell)$$

Assuming $\|\hat{\mathbf{w}}_\ell\| = \|\mathbf{w}_\ell\| + o(\|\mathbf{w}_\ell\|)$, yields

$$\|\mathbf{w}_\ell - \hat{\mathbf{w}}_\ell\|^2 = 2\|\mathbf{w}_\ell\|^2 + o(\|\mathbf{w}_\ell\|^2) - 2\|\mathbf{w}_\ell\|^2 \cos(\theta_\ell) + o(\|\mathbf{w}_\ell\|^2)$$
$$= 2\|\mathbf{w}_\ell\|^2 \cdot (1 - \cos(\theta_\ell)) + o(\|\mathbf{w}_\ell\|^2).$$

Hence, normalizing both sides by $2\|\mathbf{w}_\ell\|^2$, we obtain that

$$(1 - \cos(\theta_\ell)) = \frac{\|\mathbf{w}_\ell - \hat{\mathbf{w}}_\ell\|^2}{2\|\mathbf{w}_\ell\|^2} + o(1). \tag{12}$$

Combining the analysis of eq. (11) with eq. (12), we obtain

$$(1 - \cos(\theta_\ell)) = \frac{\|\mathbf{w}_\ell - \hat{\mathbf{w}}_\ell\|^2}{2\|\mathbf{w}_\ell\|^2} + o(1) = \frac{\Delta_\ell^2 \cdot n_\ell}{24 \cdot \|\mathbf{w}_\ell\|^2} + o(1).$$

By denoting $\epsilon_\ell = 1 - \cos(\theta_\ell)$, and omitting the little order $o(1)$, the lemma follows. □

Note that even when $R = O(\log(\|\mathbf{w}_\ell\|))$, where the error bound is relaxed to $O(1)$ still provide meaningful results. In this case, the resulting approximation error would be $O(1/\|\mathbf{w}_\ell\|)$.

## A.2 PROOF OF COROLLARY 1

*Proof.* Let $\theta_{[1:L]}$ be the rotation angle from $\mathbf{w}_{[1:L]}$ to $\hat{\mathbf{w}}_{[1:L]}$ such that $\langle \frac{\mathbf{w}_{[1:L]}}{\|\mathbf{w}_{[1:L]}\|}, \frac{\hat{\mathbf{w}}_{[1:L]}}{\|\hat{\mathbf{w}}_{[1:L]}\|} \rangle \triangleq \cos(\theta_{[1:L]})$. Assuming $\|\hat{\mathbf{w}}_\ell\| = \|\mathbf{w}_\ell\| + o(\|\mathbf{w}_\ell\|)$, then,

$$\cos(\theta_{[1:L]}) = \langle \frac{\mathbf{w}_{[1:L]}}{\|\mathbf{w}_{[1:L]}\|}, \frac{\hat{\mathbf{w}}_{[1:L]}}{\|\hat{\mathbf{w}}_{[1:L]}\|} \rangle$$

$$= \frac{\sum_{\ell=1}^{L} \langle \mathbf{w}_\ell, \hat{\mathbf{w}}_\ell \rangle}{\|\mathbf{w}_{[1:L]}\| \cdot \|\hat{\mathbf{w}}_{[1:L]}\|}$$

$$= \frac{\sum_{\ell=1}^{L} \|\mathbf{w}_\ell\| \cdot \|\hat{\mathbf{w}}_\ell\| \cdot \langle \frac{\mathbf{w}_\ell}{\|\mathbf{w}_\ell\|}, \frac{\hat{\mathbf{w}}_\ell}{\|\hat{\mathbf{w}}_\ell\|} \rangle}{\|\mathbf{w}_{[1:L]}\| \cdot \|\hat{\mathbf{w}}_{[1:L]}\|}$$

$$= \sum_{\ell=1}^{L} \frac{\|\mathbf{w}_\ell\|^2}{\|\hat{\mathbf{w}}_{[1:L]}\|^2} \cos(\theta_\ell) + o(\frac{\|\mathbf{w}_\ell\|^2}{\|\hat{\mathbf{w}}_{[1:L]}\|^2})$$

$\square$

Since $\sum_{\ell=1}^{L} \frac{\|\mathbf{w}_\ell\|^2}{\|\mathbf{w}_{[1:L]}\|^2} = 1$, the parameters' distortion is simply a convex combination of the layers' distortion.

### A.3 PROOF OF PROPOSITION 1

*Proof.* By eq. (12),

$$1 - \cos(\theta_f) = \frac{\|\hat{f}(x) - f(x)\|^2}{2 \cdot \|f(x)\|^2} + o(1).$$

where $\theta_f$ is the deviation angle obtained by eq. (1).

To make the dependence of $f(x)$ on its weights explicit, let us denote $f(x) = f_x(\mathbf{w}_{[1:L]})$ and $\hat{f}(x) = f_x(\hat{\mathbf{w}}_{[1:L]})$. Since the denominator is independent of the quantization, it is sufficient to focus on the enumerator. Accordingly, we wish to examine

$$\|f_x(\hat{\mathbf{w}}_{[1:L]}) - f_x(\mathbf{w}_{[1:L]})\|^2 = \|f_x(\mathbf{w}_{[1:L]} + \boldsymbol{\epsilon}) - f_x(\mathbf{w}_{[1:L]})\|^2 \triangleq g(\boldsymbol{\epsilon})$$

where $\boldsymbol{\epsilon}$ is (random) quantization errors. Our goal is to prove that $\mathbb{E}_{\boldsymbol{\epsilon}}[g(\boldsymbol{\epsilon})]$ is monotonically decreasing in $k$ (i.e., higher rate must reduce the quantized model deviation). Using Taylor expansion at $\boldsymbol{\epsilon} = \mathbf{0}$, we have

$$g(\boldsymbol{\epsilon}) = g(\mathbf{0}) + \nabla g(\mathbf{0}) \cdot \boldsymbol{\epsilon} + \frac{1}{2}\boldsymbol{\epsilon}^T \mathcal{H}(g(\mathbf{0}))\boldsymbol{\epsilon} + o(\boldsymbol{\epsilon}^3)$$

$$= \frac{1}{2}\boldsymbol{\epsilon}^T \mathcal{H}(g(\mathbf{0}))\boldsymbol{\epsilon}$$

where the last step follows since $g(\mathbf{0}) = \mathbf{0}$, and noting that $\nabla g(\mathbf{0}) = 0$. Finally, omitting the little order $o(\boldsymbol{\epsilon}^3)$, which is negligible in the high rate regime.

Recall that in the high rate regime the error in entry $i$, $\epsilon_i \sim U[-\Delta/2, \Delta/2]$ is i.i.d. uniformly distributed. Accordingly, in each layer $\ell$ the corresponding sub-vector $\boldsymbol{\epsilon}_\ell$ satisfies $\mathbb{E}[\boldsymbol{\epsilon}_\ell \boldsymbol{\epsilon}_\ell^T] = \Delta_\ell^2/12 \cdot \mathbf{I}_{n_\ell}$, where $\mathbf{I}_{n_\ell}$ is the $n_\ell \times n_\ell$ identity matrix. Since the errors are independent with zero mean and variance $\Delta_\ell^2/12$, we can utilize the Hutchinson (1989) trick. Accordingly, let $N = \sum_{\ell=1}^{L} n_\ell$, then

$$\frac{1}{2}\mathbb{E}_{\boldsymbol{\epsilon}}\left[\boldsymbol{\epsilon}^T \mathcal{H}(g(\mathbf{0}))\boldsymbol{\epsilon}\right] = \frac{1}{2}\mathbb{E}_{\boldsymbol{\epsilon}}\left[\sum_{i=1,j=1}^{N} \epsilon_i \mathcal{H}(g(\mathbf{0}))_{ij}\epsilon_j\right]$$

$$= \sum_{i=1,j=1}^{N} \mathcal{H}(g(\mathbf{0}))_{ij}\mathbb{E}_{\boldsymbol{\epsilon}}[\epsilon_i \epsilon_j]$$

$$= \sum_{i=1}^{N} \mathcal{H}(g(\mathbf{0}))_{ii}\mathbb{E}_{\boldsymbol{\epsilon}}[\epsilon_i^2]$$

where the last step follows since $\mathbb{E}_{\boldsymbol{\epsilon}}[\epsilon_i \cdot \epsilon_j] = 0 \forall i \neq j$. Finally, letting

$$\vec{\Delta} \triangleq \left(\frac{\Delta_1^2}{12}, \ldots, \frac{\Delta_L^2}{12}\right)^T = \frac{1}{k^2}\left(\frac{\|\mathbf{w}_1\|^2}{12}, \ldots, \frac{\|\mathbf{w}_L\|^2}{12}\right)^T,$$

we can present the last step vectorially as

$$\vec{\Delta}^T \text{diag}\left(\mathcal{H}(g(\mathbf{0}))\right) = \frac{1}{12 \cdot k^2} \cdot \left(\|\mathbf{w}_1\|^2, \ldots, \|\mathbf{w}_L\|^2\right) \cdot \text{diag}\left(\mathcal{H}(g(\mathbf{0}))\right).$$

Since $g(\mathbf{0})$ is a quadratic function with minimum at $\boldsymbol{\epsilon} = (0)$, thus $\mathcal{H}(g(\mathbf{0}))$ is positive definite, which means that its diagonal entries are real and non-negative. Further, the entries of $\vec{\Delta}$ are positive, hence, this dot product is monotonically decreasing in $k$, as $O(1/k^2)$, which completes the proof. Note that this proposition holds for any $\mathbf{x}$. $\square$

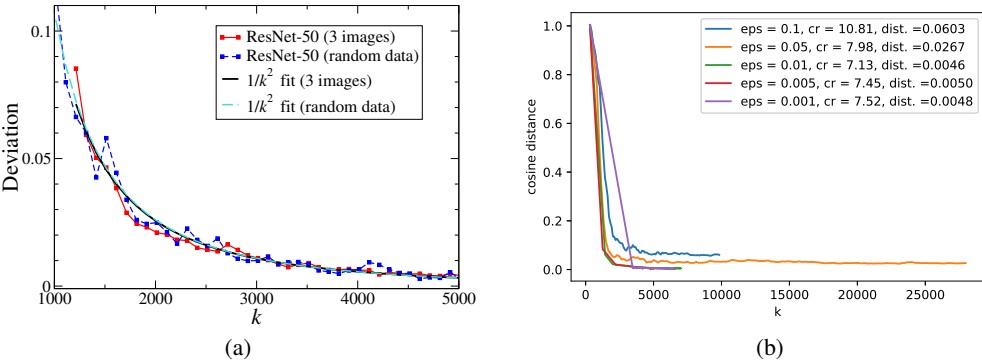

Figure 2: (a) Validation of Proposition 1 on ResNet-50. (b) The impact of $\epsilon_0$ on the performance of RIQ on ResNet50. Higher values of $\epsilon_0$ attains higher compression, yet, higher deviation.

Figure 2(a) validates the analysis of Proposition 1. In particular, we measure the deviation for ResNet-50 as a function of $k$ using both random data and a calibration set. Then, we compare the resulting deviations with a fitted curve of the form $y = a/k^2$. Clearly, the monotonicity allows to search efficiently the optimal solution.

### A.4 PROOF OF PROPOSITION 2

*Proof.* The layer whose quantization error converges last to $\epsilon_0$ dictates when to stop the search. Specifically, when $k$ is sufficiently large in eq. (7), the error in layer $\ell^*$ reaches $\sqrt{\epsilon_{\ell^*}} = o(\epsilon_0) + \epsilon_0$, where $o(\cdot)$ denotes little order of magnitude. That is where $\epsilon_0$ becomes dominant. At this point, we say that the error has converged for all layers (as it converged even at the largest layer $\ell^*$). Since $\epsilon_0 \leq 1$ in the cosine distance criterion, we choose the little order of magnitude to be $o(\epsilon_0) = \epsilon_0 \cdot \sqrt{\epsilon_0}$, and hence, $k$ can be bounded from above by

$$\frac{1}{k}\sqrt{n_{\ell^*}/24} + \epsilon_0 \geq o(\epsilon_0) + \epsilon_0,$$

which happens when $k \leq \sqrt{n_{\ell^*}/24}/(\epsilon_0 \cdot \sqrt{\epsilon_0})$.

In our experiments, we let $\epsilon_0 = 0.01$, hence, the upper limit is simply $k \leq 1000 \cdot \sqrt{n_{\ell^*}/24}$ [3]

For a lower bound, we use again the fact that $\epsilon_\ell \leq 1$. Thus, focusing on layer $\ell^*$, we observe that

$$\frac{1}{k} \cdot \sqrt{n_{\ell^*}/24} + \epsilon_0 \leq 1,$$

which happens as long as $k \geq \sqrt{n_{\ell^*}/24}/(1 - \epsilon_0)$. This completes the proof. $\square$

As mentioned, the power of $\epsilon_0$ is substantial. On the one hand, $\epsilon_0$ shifts the bin-width $\Delta_\ell$ from the exact rotation-invariant solution. On the other hand, without this term, the quantization rate

---

[3] When the original weights are represented with $R$ bit symbols, then, choosing $\epsilon_0 = 0$, yields a trivial upper bound $k_{\max}$, which is the largest number that can be represented with $R$ bits, e.g., $\lceil k_{\max} \rceil \leq 2^{31}$, when using 32 bits integer.

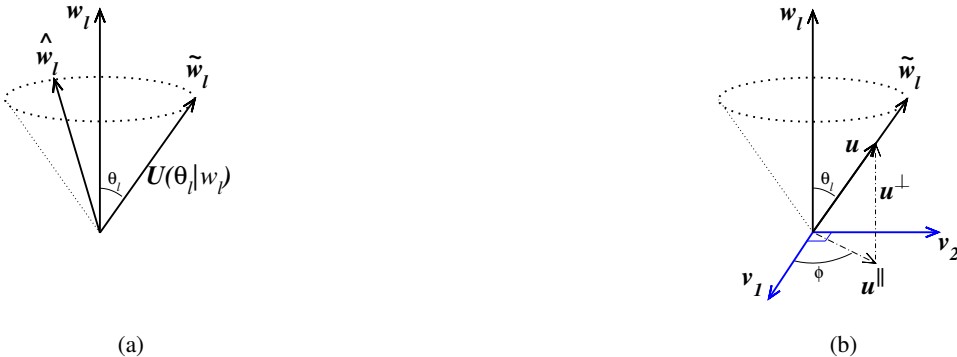

Figure 3: (a) Illustration of the surrogate model in Section 4.2. Is this illustration, the quantized weights are modeled by the result of an orthogonal transformation $\mathbf{U}(\theta_\ell|\mathbf{w}_\ell)$ which rotates the vector $\mathbf{w}_\ell$ randomly onto a ring that is $\theta_\ell$ away from $\mathbf{w}_\ell$. Note that the true quantization results also lies in this ring. (b) Illustration of the projection of $\tilde{\mathbf{w}}_\ell$ onto the arbitrary perpendicular vectors $\mathbf{v}_1$ and $\mathbf{v}_2$.

might get too high, and hence, it has practical merit. To examine its influence, we used RIQ on ResNet50 with different values of $\epsilon_0$, recording the deviation as a function of the $k$ parameter and the achieved compression (see Figure 2(b)). Interestingly, large values of $\epsilon_0$ did not reach the target deviation (which is 0.005 in our experiments). Yet, since enforces lower rates, then, it attains a good compression ratio. On the other hand, picking small $\epsilon_0$ is beneficial in terms of model deviation, which is crucial for accuracy. The best value tradeoffs between these options. That is, picking the largest value of $\epsilon_0$ that satisfies the deviation requirement.

### A.5 PROOF OF PROPOSITION 3

*Proof.* By Fang et al. (2018, Theorem 4.3), a necessary and sufficient condition for $\tilde{\mathbf{w}}_\ell|\mathbf{w}_\ell$ to have a spherical rotation-invariant distribution on a cone that is $\theta_\ell$ away from $\mathbf{w}_\ell$ is when

$$p\left(\langle\tilde{\mathbf{w}}_\ell,\mathbf{v}_1\rangle|\langle\tilde{\mathbf{w}}_\ell,\mathbf{v}_2\rangle,\mathbf{w}_\ell\right) \overset{d}{=} p\left(-\langle\tilde{\mathbf{w}}_\ell,\mathbf{v}_1\rangle|\langle\tilde{\mathbf{w}}_\ell,\mathbf{v}_2\rangle,\mathbf{w}_\ell\right),$$

for any pair of perpendicular vectors $\mathbf{v}_1 \neq 0$ and $\mathbf{v}_2 \neq 0$ that are orthogonal to $\mathbf{w}_\ell$.

Consider the model in eq. (9), any orthogonal transformation $\mathbf{U}(\theta_\ell|\mathbf{w}_\ell)$ can be represented by an orthonormal basis, obtained by the Gram-Schmidt process. That is, finding two orthonormal vectors $\mathbf{u}_1$ and $\mathbf{u}_2$ that span the plane of rotation generated by $\mathbf{w}_\ell$ and some $\mathbf{w}'_\ell$ that is $\theta_\ell$ away from $\mathbf{w}_\ell$, and then, extend this basis to $\mathbb{R}^{n_\ell}$. This allows us to consider the rotation in the plane generated by those vectors, with respect to the extended basis (https://stephenmontgomerysmith.github.io/). Accordingly, let $\mathbf{u}_1 = \frac{\mathbf{w}_\ell}{\|\mathbf{w}_\ell\|}$ and $\mathbf{u}_2 = \frac{\mathbf{w}'_\ell - \langle\mathbf{u}_1,\mathbf{w}'_\ell\rangle\mathbf{u}_1}{\|\mathbf{w}'_\ell - \langle\mathbf{u}_1,\mathbf{w}'_\ell\rangle\mathbf{u}_1\|}$, then

$$\mathbf{U}(\theta_\ell|\mathbf{w}_\ell) = \mathbf{I}_{n_\ell} - \mathbf{u}_1\mathbf{u}_1^T - \mathbf{u}_2\mathbf{u}_2^T + [\mathbf{u}_1,\mathbf{u}_2]\,\mathbf{R}_{\theta_\ell}\,[\mathbf{u}_1,\mathbf{u}_2]^T, \tag{13}$$

where $\mathbf{I}_{n_\ell}$ is the $n_\ell \times n_\ell$ identity matrix and $\mathbf{R}_{\theta_\ell}$ is the rotation matrix

$$\mathbf{R}_{\theta_\ell} = \left[\begin{array}{cc} \cos(\theta_\ell) & -\sin(\theta_\ell) \\ \sin(\theta_\ell) & \cos(\theta_\ell) \end{array}\right],$$

that rotates at a scalar angle $\theta_\ell$, and $[\mathbf{u}_1,\mathbf{u}_2]$ is $n_\ell \times 2$ matrix whose columns are $\mathbf{u}_1$ and $\mathbf{u}_2$, respectively. Plugging eq. (13) to eq. (9), and noting that $\frac{\mathbf{w}_\ell}{\|\mathbf{w}_\ell\|} = \mathbf{u}_1$, we obtain

$$\tilde{\mathbf{w}}_\ell = \|\hat{\mathbf{w}}_\ell\| \cdot (\cos(\theta_\ell)\mathbf{u}_1 + \sin(\theta_\ell)\mathbf{u}_2) \tag{14}$$

To simplify notation, let $\mathbf{u} \triangleq (\cos(\theta_\ell)\mathbf{u}_1 + \sin(\theta_\ell)\mathbf{u}_2)$, and note that for any perpendicular pair $\mathbf{v}_1, \mathbf{v}_2$ that are orthogonal to $\mathbf{w}_\ell$, the vector $\mathbf{u}$ can be decomposed to $\mathbf{u} = \mathbf{u}^\| + \mathbf{u}^\perp$, where $\mathbf{u}^\|$ resides in the plane generated by $\mathbf{v}_1$ and $\mathbf{v}_2$, and $\mathbf{u}^\perp$ resides in the null-space of this plane. For illustration, see Figure 3(b). Hence,

$$\tilde{\mathbf{w}}_\ell = \|\hat{\mathbf{w}}_\ell\| \cdot (\mathbf{u}^\| + \mathbf{u}^\perp) \tag{15}$$

Accordingly, we have

$$p\left(\langle \tilde{\mathbf{w}}_\ell, \mathbf{v}_1 \rangle \middle| \langle \tilde{\mathbf{w}}_\ell, \mathbf{v}_2 \rangle, \mathbf{w}_\ell \right) \tag{16}$$

$$\stackrel{(a)}{=} p\left( \langle \|\hat{\mathbf{w}}_\ell\| \cdot (\mathbf{u}^\| + \mathbf{u}^\perp), \mathbf{v}_1 \rangle \middle| \langle \|\hat{\mathbf{w}}_\ell\| \cdot (\mathbf{u}^\| + \mathbf{u}^\perp), \mathbf{v}_2 \rangle, \mathbf{w}_\ell \right)$$

$$\stackrel{(b)}{=} p\left( \langle \|\hat{\mathbf{w}}_\ell\| \cdot \mathbf{u}^\|, \mathbf{v}_1 \rangle \middle| \langle \|\hat{\mathbf{w}}_\ell\| \cdot \mathbf{u}^\|, \mathbf{v}_2 \rangle, \mathbf{w}_\ell \right)$$

$$\stackrel{(c)}{=} p\left( \|\hat{\mathbf{w}}_\ell\| \cdot \|\mathbf{u}^\|\| \cdot \cos(\phi) \middle| \|\hat{\mathbf{w}}_\ell\| \cdot \|\mathbf{u}^\|\| \cdot \sin(\phi), \mathbf{w}_\ell \right)$$

$$\stackrel{(d)}{=} p\left( \|\hat{\mathbf{w}}_\ell\| \cdot \|\mathbf{u}^\|\| \cdot \cos(\pi - \phi) \middle| \|\hat{\mathbf{w}}_\ell\| \cdot \|\mathbf{u}^\|\| \cdot \sin(\pi - \phi), \mathbf{w}_\ell \right)$$

$$\stackrel{(e)}{=} p\left( -\|\hat{\mathbf{w}}_\ell\| \cdot \|\mathbf{u}^\|\| \cdot \cos(\phi) \middle| \|\hat{\mathbf{w}}_\ell\| \cdot \|\mathbf{u}^\|\| \cdot \sin(\phi), \mathbf{w}_\ell \right)$$

$$= p\left( -\langle \hat{\mathbf{w}}_\ell, \mathbf{v}_1 \rangle \middle| \langle \hat{\mathbf{w}}_\ell, \mathbf{v}_2 \rangle, \mathbf{w}_\ell \right) \tag{17}$$

where (a) follows by eq. (15). (b) follows by the linearity of the inner product and since $\mathbf{u}^\perp$ is perpendicular to both $\mathbf{v}_1$ and $\mathbf{v}_2$ (c) follows due to the orthogonality of the basis $\mathbf{v}_1$ and $\mathbf{v}_2$, where $\phi$ is the angle between $\mathbf{u}^\|$ and $\mathbf{v}_1$. (d) follows since the angle between $\mathbf{u}^\|$ and an arbitrary $\mathbf{v}_1$ is arbitrary, and hence, every angle has the same distribution. (e) follows by trigonometric identities for the cosine and sine function.

Thus, Proposition 3 follows. $\qquad\square$

### A.6 PROOF OF THEOREM 1

*Proof.* Consider the rate-distortion function

$$R(D) = \min_{\substack{p\left(\tilde{\mathbf{w}}_{[1:L]} \middle| \mathbf{w}_{[1:L]}\right): \\ \mathbb{E}\left[d_{f,x}(\mathbf{w}_{[1:L]}, \tilde{\mathbf{w}}_{[1:L]})\right] \leq D}} I\left(\mathbf{w}_{[1:L]}; \tilde{\mathbf{w}}_{[1:L]}\right)$$

Assuming the weights of each layer $\mathbf{w}_\ell$ are statistically independent of the weights of the other layers, then, by the properties of the mutual information, we have

$$I\left(\mathbf{w}_{[1:L]}; \tilde{\mathbf{w}}_{[1:L]}\right) = H\left(\mathbf{w}_{[1:L]}\right) - H\left(\mathbf{w}_{[1:L]} \middle| \tilde{\mathbf{w}}_{[1:L]}\right) \tag{18}$$

$$= \sum_{\ell=1}^{L} H\left(\mathbf{w}_\ell\right) - \sum_{\ell=1}^{L} H\left(\mathbf{w}_\ell \middle| \mathbf{w}_{[1:\ell-1]}, \tilde{\mathbf{w}}_{[1:L]}\right) \tag{19}$$

$$\geq \sum_{\ell=1}^{L} H\left(\mathbf{w}_\ell\right) - \sum_{\ell=1}^{L} H\left(\mathbf{w}_\ell \middle| \tilde{\mathbf{w}}_\ell\right) \tag{20}$$

$$= \sum_{\ell=1}^{L} I\left(\mathbf{w}_\ell; \tilde{\mathbf{w}}_\ell\right) \tag{21}$$

$$\geq \sum_{\ell=1}^{L} R(D_\ell) \tag{22}$$

where eq. (20) follows since conditioning reduces entropy. Note, however, that eq. (20) can be attained with equality by letting $p\left(\mathbf{w}_{[1:L]} \middle| \tilde{\mathbf{w}}_{[1:L]}\right) = \prod_{\ell=1}^{L} p\left(\mathbf{w}_\ell \middle| \tilde{\mathbf{w}}_\ell\right)$. Consequently, the minimizing distribution in eq. (10) is a product distribution (Polyanskiy & Wu, 2022, Theorem 6.1 (2)).

Interestingly, eq. (22) implies that the optimal solution has a simple form of a *layer-by-layer solution*, which significantly simplifies the problem at hand. Finding the solution, of course, requires formulating the relation between the deviations $D$ and $D_\ell$ of each layer $\ell$, and hence, the resulting rate $R(D_\ell)$ for each layer. Moreover, since each layer obtains a different rate, it implies that the optimal solution is indeed a *mixed-precision solution*, where each layer can be considered independently, and hence, the minimizing distribution is product distribution.

First, let us consider the deviation. Let $p\left(\tilde{\mathbf{w}}_1 \middle| \mathbf{w}_1\right) \cdot p\left(\tilde{\mathbf{w}}_2 \middle| \mathbf{w}_2\right) \cdots p\left(\tilde{\mathbf{w}}_L \middle| \mathbf{w}_L\right)$ be a distribution that satisfies the deviation requirement $D$, for which the induced cosine distance (distortion) in each layer

$\ell$ is at most $\epsilon_\ell$, for $\epsilon_\ell = 1 - \cos(\theta_\ell)$. By Corollary 1, the distortion over the entire parameters is hence $\epsilon_{[1:L]} \triangleq 1 - \cos(\theta_{[1:L]}) = \sum_{\ell=1}^{L} \frac{\|\mathbf{w}_\ell\|^2}{\|\mathbf{w}_{[1:L]}\|^2} \epsilon_\ell$. Note that $\epsilon_{[1:L]}$ is a convex combination of $\epsilon_\ell$ as $\sum_{\ell=1}^{L} \frac{\|\mathbf{w}_\ell\|^2}{\|\mathbf{w}_{[1:L]}\|^2} = 1$. Assuming $\epsilon_{[1:L]} \le D$, then due to the convexity of the cosine distance for $|\theta_\ell| \le \pi/2$, by Jensen inequality $\left(1 - \cos\left(\sum_{\ell=1}^{L} \frac{\|\mathbf{w}_\ell\|^2}{\|\mathbf{w}_{[1:L]}\|^2} \theta_\ell\right)\right) \le \sum_{\ell=1}^{L} \frac{\|\mathbf{w}_\ell\|^2}{\|\mathbf{w}_{[1:L]}\|^2} (1 - \cos(\theta_\ell))$. In words, convex combination of the angles also satisfies the deviation constraint $D$. Generally, cosine distance is a rotation-invariant distance, as the angle between vectors does not change when they are rotated together. This further hints that the minimizing distribution should also be a rotation-invariant distribution as follows.

Next, let us address $p(\tilde{\mathbf{w}}_\ell|\mathbf{w}_\ell)$. By eq. (14), given $\mathbf{w}_\ell$ (and hence, $\mathbf{u}_1$), the probability of $\tilde{\mathbf{w}}_\ell$ is determined by the probability of the rotation angle $\theta_\ell$ and the norm $\|\hat{\mathbf{w}}_\ell\|$. Specifically, for any vector $\mathbf{s}_\ell \in \mathbb{R}^{n_\ell}$, the density function of this product, if exists, is (Melvin Dale, 1979, Ch. 4.1)

$$p_{\tilde{\mathbf{w}}_\ell|\mathbf{w}_\ell}(\mathbf{s}_\ell) = \int_0^\infty p_{\|\hat{\mathbf{w}}_\ell\||\mathbf{w}_\ell}(h) \cdot p_{(\cos(\theta_\ell)\mathbf{u}_1+\sin(\theta_\ell)\mathbf{u}_2)|\mathbf{w}_\ell}(\mathbf{s}_\ell/h) \cdot \frac{1}{h}\mathrm{d}h, \tag{23}$$

where the rotation $\theta_\ell$ occurs on $\mathbb{R}^2$, rotating about $(n_\ell - 2)$-dimensional subspace. Further, note that the dimension $n_\ell$ is dictated only by the given $\mathbf{w}_\ell$. Apparently, since each layer $\ell$ resides at a different dimension $n_\ell$, it is impossible to consider the convex combination of the layers' distribution directly, as done for the vector case, e.g., as considered in Polyanskiy & Wu (2016, Ch. 5). Nevertheless, since the rotation of $\theta_\ell$ is done on $\mathbb{R}^2$ in each layer $\ell$, which is described by the rotation matrix $\mathbf{R}_{\theta_\ell}$ in eq. (13), it is still beneficial to consider a convex combination of $\theta_\ell$ distributions over the layers, to allow a similar treatment to Polyanskiy & Wu (2016, Ch. 5), as follows.

To bound the mutual information, the density of $(\cos(\theta_\ell)\mathbf{u}_1 + \sin(\theta_\ell)\mathbf{u}_2)|\mathbf{w}_\ell$ should be expressed first in terms of the density of $\cos(\theta_\ell)|\mathbf{w}_\ell$. Examining eq. (14), we note that by the transformation of random variables formula,

$$p_{(\cos(\theta_\ell)\mathbf{u}_1+\sin(\theta_\ell)\mathbf{u}_2)|\mathbf{w}_\ell}(\mathbf{s}_\ell/h) = p_{\cos(\theta_\ell)|\mathbf{w}_\ell}\left(\mathbf{u}_1^T\mathbf{s}_\ell/h\right). \tag{24}$$

Considering the high rate regime, where $\Delta_\ell$ is sufficiently small, and thus, $\|\hat{\mathbf{w}}_\ell\| = \|\mathbf{w}_\ell\| + o(1)$, then, the density function in eq. (23) becomes

$$p_{\tilde{\mathbf{w}}_\ell|\mathbf{w}_\ell}(\mathbf{s}_\ell) = p_{\|\hat{\mathbf{w}}_\ell\|\cdot(\cos(\theta_\ell)\mathbf{u}_1+\sin(\theta_\ell)\mathbf{u}_2)|\mathbf{w}_\ell}(\mathbf{s}_\ell) \tag{25}$$

$$= \int_0^\infty p_{\|\hat{\mathbf{w}}_\ell\||\mathbf{w}_\ell}(h) \cdot p_{(\cos(\theta_\ell)\mathbf{u}_1+\sin(\theta_\ell)\mathbf{u}_2)|\mathbf{w}_\ell}(\mathbf{s}_\ell/h) \cdot \frac{1}{h}\mathrm{d}h \tag{26}$$

$$\overset{(a)}{=} \int_0^\infty p_{\|\hat{\mathbf{w}}_\ell\|\|\mathbf{w}_\ell\|}(h) \cdot p_{(\cos(\theta_\ell)\mathbf{u}_1+\sin(\theta_\ell)\mathbf{u}_2)|\mathbf{w}_\ell}(\mathbf{s}_\ell/h) \cdot \frac{1}{h}\mathrm{d}h \tag{27}$$

$$\overset{(b)}{\approx} \int_0^\infty \delta(h - \|\mathbf{w}_\ell\|) \cdot p_{(\cos(\theta_\ell)\mathbf{u}_1+\sin(\theta_\ell)\mathbf{u}_2)|\mathbf{w}_\ell}(\mathbf{s}_\ell/h) \cdot \frac{1}{h}\mathrm{d}h \tag{28}$$

$$\overset{(c)}{=} \int_0^\infty \delta(h - \|\mathbf{w}_\ell\|) \cdot p_{\cos(\theta_\ell)|\mathbf{w}_\ell}\left(\mathbf{u}_1^T\mathbf{s}_\ell/h\right) \cdot \frac{1}{h}\mathrm{d}h \tag{29}$$

$$\overset{(d)}{=} p_{\cos(\theta_\ell)|\mathbf{w}_\ell}\left(\mathbf{u}_1^T\mathbf{s}_\ell/\|\mathbf{w}_\ell\|\right) \cdot \|\mathbf{w}_\ell\|^{-1} \tag{30}$$

where (a) follows since the norm $\|\mathbf{w}_\ell\|$ is a function of the given $\mathbf{w}_\ell$. (b) follows since the uncertainty about $\|\hat{\mathbf{w}}_\ell\|$ given $\|\mathbf{w}_\ell\|$ is negligible, and hence, $p_{\|\hat{\mathbf{w}}_\ell\|\|\mathbf{w}_\ell\|}(h) \approx \delta(h - \|\mathbf{w}_\ell\|)$, i.e., the conditional density is approximately the Dirac delta function. (c) follows by eq. (24). (d) follows by the characteristics of the Dirac delta function.

Hence, it is possible to consider a convex combination of $p\left(\tilde{\mathbf{w}}_\ell|\mathbf{w}_\ell\right)$ over the layers, where each layer is embedded in possibly different $n_\ell$, by considering a convex combination of the rotations' probability $p_{\cos(\theta_\ell)|\mathbf{w}_\ell}$, since all rotations are done in $\mathbb{R}^2$. Accordingly, let

$$\bar{p}_{\tilde{\mathbf{w}}_{[1:L]}|\mathbf{w}_{[1:L]}}(\mathbf{s}_{[1:L]}) \triangleq \sum_{\ell=1}^{L} \frac{\|\mathbf{w}_\ell\|^2}{\|\mathbf{w}_{[1:L]}\|^2} \cdot p_{\cos(\theta_\ell)|\mathbf{w}_\ell}\left(\frac{\mathbf{w}_\ell^T\mathbf{s}_\ell}{\|\mathbf{w}_\ell\|^2}\right) \cdot \|\mathbf{w}_\ell\|^{-1}. \tag{31}$$

Then, by the convexity of the rate-distortion function Cover & Thomas (2006, Theorem 2.7.4), $\bar{p}\left(\tilde{\mathbf{w}}_{[1:L]}\big|\mathbf{w}_{[1:L]}\right)$ can only reduce the mutual information in eq. (21). Specifically,

$$\sum_{\ell=1}^{L} I_{p\left(\tilde{\mathbf{w}}_{\ell}\big|\mathbf{w}_{\ell}\right)}\left(\mathbf{w}_{\ell};\tilde{\mathbf{w}}_{\ell}\right) \geq L \cdot I_{\bar{p}\left(\tilde{\mathbf{w}}_{[1:L]}\big|\mathbf{w}_{[1:L]}\right)}\left(\mathbf{w}_{[1:L]};\tilde{\mathbf{w}}_{[1:L]}\right)$$

where $I_p(\cdot;\cdot)$ denotes explicitly the mutual information under probability $p$. Thus, the *infimum rate has a form of a scalar (single-letter) rate*.

Moreover, since averaging over more rotations should further reduce the mutual information by its convexity, then, the minimizing $p\left(\tilde{\mathbf{w}}_{[1:L]}\big|\mathbf{w}_{[1:L]}\right)$ can be chosen to be rotation-invariant (Polyanskiy & Wu, 2022, Ch. 6.2). Consequently, the unique minimizing distribution $p\left(\tilde{\mathbf{w}}_{[1:L]}\right)$ is also rotation-invariant. Remarkably, Fang et al. (2018, Theorem 2.6) states that when partitioning a spherical rotation-invariant distribution (naturally, according to the layers $\tilde{\mathbf{w}}_{\ell}$), then its components also have a spherical rotation-invariant distribution. This coincides with Proposition 3, which proves that the partitioning satisfies this property.

Accordingly, the unique minimizing distribution $p(\tilde{\mathbf{w}}_{[1:L]}\big|\mathbf{w}_{[1:L]})$ of the rate-distortion function is a product distribution over the layers, where each term $\ell$ is a spherical rotation-invariant distribution. This completes the proof.

$\square$

## A.7 RELATION TO OTHER ERROR CRITERIA

**Remark.** *The proof of Lemma 1 in Appendix A.1 may serve as a proxy to other error criteria such as the* Signal to Quantization Noise Ratio *(SQNR), Caffarena et al. (2010). Specifically, similar to the proof of Lemma 1, the resulting connection between the scale $\Delta_\ell$ and the SQNR $\epsilon'_\ell$ in each layer $\ell$ is*

$$\epsilon'_\ell \triangleq \frac{\|\boldsymbol{w}_\ell - \hat{\boldsymbol{w}}_\ell\|}{\|\boldsymbol{w}_\ell\|} = \sqrt{\frac{\Delta_\ell^2}{12} \cdot \frac{n_\ell}{\|\boldsymbol{w}_\ell\|^2}}$$

*Or, equivalently,*

$$\Delta_\ell = \epsilon'_\ell \|\boldsymbol{w}_\ell\| \sqrt{12/n_\ell}$$

## A.8 ADDITIONAL RESULTS

### A.8.1 DECOMPOSING THE RATE-DISTORTION CURVE

The key steps of lossy compression are quantization and compression. In the quantization phase, the RIQ approach is minimizing the overall model's entropy by allocating a small number of unique symbols for large-norm layers. To achieve (asymptotically) this entropy limit, we utilize the ANS (lossless) entropy encoder. In this section, we evaluate the contribution of each step to the rate-distortion tradeoff. Namely, the average rate per (quantized) symbol before and after ANS. At run-time, when a certain layer is required, it is decoded and represented at a rate according to RIQ. If this rate is below 8 bits/symbol, it enables significant acceleration by performing 8 bits integer operations, as discussed in Appendix A.8.2.

Figure 4(a) depicts the rate-distortion curve for ResNet-50, decomposed to the quantization step (dashed lines) and the resulting compression step, following the quantization step (solid lines). As a baseline, the uniform scalar quantization (red color) is given for comparison with RIQ (green color). Interestingly, RIQ (dashed green line) outperforms the uniform quantization (dashed red line) by about $\sim 4$ bits/symbol and even its resulting compressed size by about $\sim 1$ bit/symbol. Indeed, the latter indicates that uniform quantization does not minimize the model's entropy. Applying the ANS compression following RIQ reduces additional $\sim 3$ bits/symbol (solid green line), which according to our analysis is the minimum entropy possible for a given distortion. Moreover, our method achieves a reduction of about $\sim 8$ bits/symbol compared to uniform scalar quantization alone, and an additional $\sim 4$ bits/symbol when ANS is applied to the uniformly quantized weights. For completeness, in Figure 4(b), we depict the rate per layer statistics for the ResNet-50 model with a deviation constraint of $0.5\%$, using $\epsilon_0 = 0.01$. Noticeably, most of the layers require less than 8 bits. Moreover, the

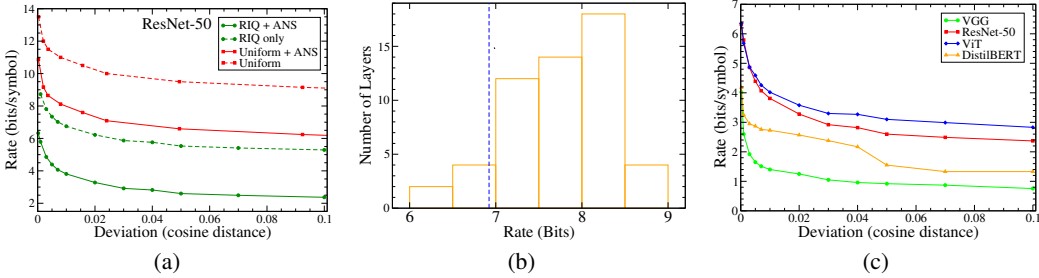

Figure 4: (a) Rate-distortion curve for ResNet-50 model obtained for RIQ (green circles) as well as Uniform linear quantization (red squares). Rates are presented for both the quantized model (dashed) as well as following an ANS compression. (b) ResNet-50 rate per layer statistics with $\epsilon_0 = 0.01$ in all layers. (c) Rate distortion curves obtained by RIQ + ANS, for a variety of models: VGG (green circles), ResNet-50 (red squares), ViT (blue diamonds), and DistilBERT (orange triangles).

average rate is 6.9 bits per symbol, as larger layers get quantized more aggressively. Hence, those layers' run time can be accelerated significantly when their input is quantized as well. Yet, if one wishes to enforce a rate that is less than 8 bit to all layers, it should pick an $\epsilon_0$ according to Section 4.1.

The rate-distortion curves for various models, in particular, the VGG (green circles), ResNet-50 (red squares), ViT (blue diamonds), and DistilBERT (orange triangles) are given in Figure 4(c). As expected, the curves decrease monotonously, reaching an impressive compression rate of less than 8 bits/symbol on average even for extremely low cosine distance in all presented models. Table 3 depicts the compression ratio attained for various models[4] and tasks, with 1% deviation constraint. Interestingly, from this table, we can infer that the MobileNet-v2 and ArcFace models are relatively efficient since their compression ratio is lower than the other (over-parameterized) models.

One of the strengths of RIQ is its efficiency. In fact, for ResNet-50 (whose size is 102 MB), it takes less than 1 minute to find $k$ and compress the model on a CPU. Note, however, that finding the optimal $k$ is typically an offline procedure, and thus it is not time critical.

Table 3: Compression and deviation attained by RIQ for various models and tasks under deviation constraint of 1%.

| Model | Deviation constraint | Compression ratio |
|---|---|---|
| MobileNet-v2 (image classification) | 1 % | ×5.79 |
| ArcFace (face detection) | 1 % | ×7.75 |
| AlexNet (image classification) | 1 % | ×19.3 |
| Candy (style transfer) | 1 % | ×20.8 |
| AgeGoogleNet (gender and age) | 1 % | ×26.95 |

Table 4 describes the performance of RIQ on large language models. In particular, we used RIQ to compress the llama-7b ×4, and then evaluated the quantized model using Gao et al. (2021).

Table 4: Llama-7b performance with ×4 compression.

| Task | Metric | Baseline | RIQ |
|---|---|---|---|
| ARC challenge | acc (%) | 41.89 % | 41.98% |
| SWAG | acc norm (%) | 76.60% | 75.95% |
| XWinograd | acc (%) | 87.87 % | 87.18% |
| piqa | acc (%) | 78.67% | 77.97% |

Table 5 compares the accuracy and compression ratio attained by RIQ with various baseline techniques. This tables shows that creating a smaller model by knowledge distillation attains impressive accuracy

---

[4] github.com/onnx/models

(sometimes even higher than the baseline), yet, falls short in terms of compression ratio. Other baselines that quantizes both the activations and weights suffer from high accuracy drop.

Table 5: Comparison of RIQ with various baseline compression techniques on various tasks.

| Model | Method | Comp. | Acc. (%) | Ref. (%) | Drop (%) |
|---|---|---|---|---|---|
| ViT I1k | MiniViT (KD) Zhang et al. (2022b) | ×2 | **84.7** | **77.9** | **-6.8** |
|  | **RIQ (Ours)** | × **6.98** | 81.0 | 81.07 | 0.07 |
| YOLO/COCO | DRGS Wu et al. (2022) | ×8 | 33.4 | 55.7 | 22.3 |
|  | **RIQ (Ours)** | ×**8.34** | **54.7** | **55.7** | **1.0** |
| DistilBERT/SQuAD | OFA (KD) Zafrir et al. (2021) | ×6.67 | **88.82** | **85.8** | **-0.02** |
|  | **RIQ (Ours)** | × **7.96** | 85.0 | 85.08 | 0.08 |

### A.8.2 ROTATION INVARIANT QUANTIZATION WITH QUANTIZED ACTIVATIONS

Quantizing both the NN model's weights and its activations can further accelerate the inference, Wu et al. (2020); Nagel et al. (2020); Krishnamoorthi (2018). Nevertheless, in this case, the quantization error of both the weights and the activation affects the model's output. In the seminal work of Wu et al. (2020), the authors utilized the KL distance for quantizing the activations to minimize the information loss at the output. In this section, we examine the RIQ approach, combining it with activation quantization.

To demonstrate, we use the NVIDIA (2021) quantization library for the ResNet-50 model with a "mini ImageNet" validation set, which comprises one image per class and a total of 1000 images. We evaluate this library's performance as a baseline, where the activations are quantized by the KL-distance criterion, and the weights are quantized to 8-bit linearly. The resulting cosine distance at the output of this baseline is 0.69%. For comparison, this reference value is given as the deviation requirement to RIQ. In particular, to integrate RIQ, the activations are quantized as the baseline, and then, we run RIQ according to Algorithm 1. This way, RIQ is aware of the activations' quantization error during its search for the single-letter solution. Note that to facilitate the acceleration of int8 operations, RIQ must yield a quantization rate of up to 8 bits/symbol. In case a certain layer requires a higher rate, we simply perform linear uniform quantization to 8 bits (without clipping), as the baseline does.

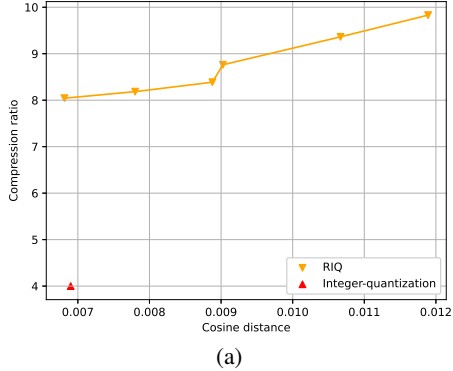
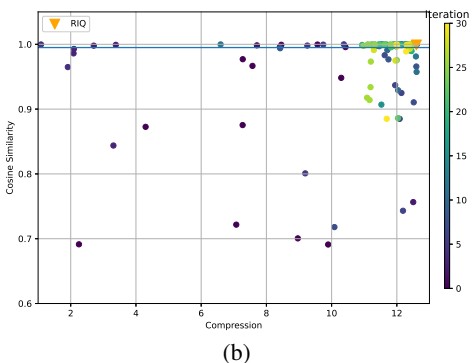

(a)          (b)

Figure 5: (a) The compression ratio as a function of cosine distance. The left-bottom red triangle depicts the resulting distance of 0.0069 achieved by the baseline with a compression ratio of ×4. The orange upside-down triangles depict the cosine distance and compression ratio attained by RIQ with the ANS compression. The orange line depicts the trend line. (b) MOBO optimization process. Interestingly, MOBO converges at the few last iterations to ×12 compression, with a highest value of ×12.61. On the other hand, RIQ reaches practically the same compression ratio in a few seconds.

Figure 5(a) characterizes the compression ratio as a function of cosine distance. The leftmost point reflects a cosine distance of 0.0069 achieved by the baseline of NVIDIA (2021). Remarkably, the RIQ attains superior compression with relatively low deviation even when the activations are quantized.

In run-time, of course, the reconstructed values are represented again by 8-bit value, and hence, the significant acceleration of Wu et al. (2020) is still valid.

### A.8.3 COMPARISON WITH MULTI-OBJECTIVE BAYESIAN OPTIMIZATION

In this section, we utilize the *Multi-Objective Bayesian Optimization* (MOBO) tool, described in Daulton et al. (2020) to compress NN models, and compare results with RIQ. To compress models with MOBO, we set two objective functions for it. The first objective is minimizing the cosine distance in eq. (1). The second objective is maximizing the compression ratio in eq. (2). Then, we let MOBO optimize the rate-distortion tradeoff (i.e., the Pareto frontier surface).

Nonetheless, MOBO is quite complex and requires strong computing capabilities for exploration and exploitation. Particularly, reaching the optimal solution may take days and even weeks, using multiple GPUs. Even on small NN models, to address the high-dimensional search spaces, we apply sparse axis-aligned subspace priors for Bayesian optimization (qNEHVI + SAASBO), with the batch Noisy Expected Improvement (qNEI) acquisition function, as suggested by Eriksson & Jankowiak (2021); Daulton et al. (2021b;a). Moreover, since the two objectives are not within the same range the cosine similarity objective had to be scaled accordingly to converge to the optimal solution, where a calibration set of 4 images are used during 30 iterations of exploration/exploitation.

Accordingly, we pick a (relatively) small model for comparison (with a size of 112 KB), letting MOBO to find for each layer its optimal bin width and quantize accordingly. We emphasize that the MOBO solution does not rely on the rotation invariant insights. In Figure 5(b), the optimization process of MOBO is presented, where each dot depicts experiment results, and its color indicates the iteration in which this result was attained. The compression results of RIQ are presented for comparison. Remarkably, RIQ and MOBO attained almost identical results of $\times 12.6$ and $\times 12.61$, respectively, with a cosine distance of $0.005$. This indicates that RIQ reaches the optimal solution.

