# OpenReview forum: "Rotation Invariant Quantization for Model Compression"
_ICLR.cc/2024/Conference — Submitted to ICLR 2024_

### Official Review · Reviewer_qJtZ · 2023-10-30

**Soundness:** 1 poor
**Presentation:** 3 good
**Contribution:** 2 fair
**Rating:** 3
**Confidence:** 4

**Summary:**

The paper examines post-training quantization of neural networks featuring linear layers, taking into account cosine distortion. The authors first demonstrate the rate-distortion trade-off between the original and quantized weights to establish the step size, $\Delta_{\ell}$, for each layer. Notably, all $\Delta_{\ell}$ values are governed by a singular parameter, $k$. Additionally, the paper investigates the rate-deviation analysis, wherein the deviation assesses the disparity in output. The authors introduce surrogate models in which the quantized weight is uniformly distributed across a space subject to random rotation, characterized by the angle $\theta_{\ell}$. Subsequently, it is proven that the mutual information is minimized when using the product distribution.

**Strengths:**

1. The paper is well-written and straightforward to understand.
2. The concept of rotational invariance in neural network quantization is intriguing.

**Weaknesses:**

1. The central assumption, $\||w\|| = \||\hat{w}\|| + o(\||w\||)$, appears to be contentious. For instance, with fixed-bit quantization, the discrepancy between $\||w\||$ and $\||\hat{w}\||$ is proportional to $\||w\||$, or in other words, $O(\||w\||)$.

2. Lemma 1 could benefit from a more rigorous presentation, especially concerning the $o(\cdot)$. Additionally, both Lemma 1 and Corollary 1 do not appear to offer significant novel contributions.

3. The surrogate model feels somewhat contrived. Moreover, the method for deriving $\tilde{w}_{\ell}$ is not clear. For instance, are the angles $\theta_{\ell}$ specified? The notion of being "uniformly distributed on a cone" is ambiguous due to its unbounded norm. As a result, Theorem 1 requires a more comprehensive problem definition.

4. There seem to be some technical inaccuracies in the proof of Theorem 1. It is also essential to ensure that each mutual information,
$I(w_{\ell}, \tilde{w}_{\ell})$, adheres to the distortion criteria.

5. The link between distortion (related to weights) and deviation (pertaining to output) is unclear.

**Questions:**

Please check Weakness.

---

> ### Author Response · Authors · 2023-11-14
>
> *We would like to thank the reviewer for the time and effort of the thorough review and helpful comments!*
>
> - The central assumption, $\Vert w \Vert = \Vert \hat{w}\Vert + o(\Vert w \Vert )$, appears to be contentious. For instance, with fixed-bit quantization, the discrepancy between and is proportional to $O(\Vert w \Vert)$
>
> [A]  The order of discrepancy depends on the rate regime that is considered. This paper considers the high-rate regime as mentioned explicitly throughout the paper.
>
> As the reviewer pointed out, the quantization error in each layer is bounded by $\Vert \epsilon \Vert \leq \sqrt{n}\frac{\Delta}{2}$. Assuming that the range of $w$ is symmetric around zero for simplicity, then, $\Delta \approx \frac{2 max(w)}{2^R}$, where $R$ is the rate of quantization. Hence, the error $\Vert \epsilon\Vert \leq \sqrt{n}\frac{max(w)}{2^R} \approx \frac{\Vert w \Vert}{2^R}$, and thus, the $o(\cdot)$ is contentious.
>
> Yet, $O(\Vert w\Vert)$ does not reflect the high rate regime that is considered in the paper. In particular, when letting $R = O(\log^c(\Vert w \Vert))$ for $c>1$, then, we obtain the desired $o(\Vert w \Vert)$.
> Moreover, even when $R = O(\log(\Vert w \Vert))$, which results in a relaxed error of $O(1)$, still provides meaningful results. In this case, the resulting approximation error would be $O(1/\Vert w \Vert)$.
>
> We thank the reviewer for pointing this out and agree that the discussion above would improve the presentation and clarity of the lemma. In the revised version, we explicitly restrict the lemma to the high-rate regime, and we provide the above discussion in the proof of Lemma 1.
>
> - Lemma 1 could benefit from a more rigorous presentation, especially concerning the $o(\cdot)$. Additionally, both Lemma 1 and Corollary 1 do not appear to offer significant novel contributions.
>
> [A] As mentioned, in the revised version of Lemma 1 proof, we provide the above discussion about the high rate regime, which we believe should address the concern about $o(\cdot)$. Please note that the goal of Lemma 1 is to provide the asymptotic connection between the bin width and the resulting cosine distance in the high-rate regime. As far as we know, this connection is missing in the literature.
>
> Corollary 1 connects the overall cosine distance to the layers' cosine distance, and it is used in the proof of Theorem 1, where we used this convex combination representation to minimize the mutual information. Furthermore, this corollary connects the distortion of the layers to the distortion when considering the whole parameters.
>
> - The surrogate model feels somewhat contrived.  Moreover, the method for deriving $\tilde w_\ell$ is not clear. For instance, are the angles $\theta_\ell$ specified? The notion of being "uniformly distributed on a cone" is ambiguous due to its unbounded norm. As a result, Theorem 1 requires a more comprehensive problem definition.
>
> [A]  Note that $\hat w_\ell$, and hence, its norm and  $\theta_\ell$ are specified by the quantization. The $\tilde w_\ell$ models the quantized vector $\hat w_\ell$. In particular, the surrogate model considers all the vectors that are $\theta_\ell$ away by using a random rotation matrix $U(\theta_\ell)$ (see Figure 3 for illustration). That is, $\tilde w_\ell$ is a realization of this random rotation. Note that the norm of $\tilde w_\ell$ is the same as $\hat w_\ell$ (and hence, bounded). Thus, any realization of  $\tilde w_\ell$ is uniformly distributed on the cone's ring.
>
> In the revised version we clarify this point after introducing the surrogate model.
>
> - There seem to be some technical inaccuracies in the proof of Theorem 1. It is also essential to ensure that each mutual information $I(w_\ell,  \tilde w_\ell)$ adheres to the distortion criteria.
>
> [A] As mentioned in the problem statement, our goal is to minimize the cosine distance between the models' output. In the rate-distortion analysis, we focus on solutions in which the cosine distance between the outputs is small, and not in the cosine distance per each layer. In other words, we are interested in the total deviation. A good analogy would be vector quantization vs. scalar quantization, where restricting the quantization error in each element would lose optimality (See e.g., [Ch. 24.2](https://people.lids.mit.edu/yp/homepage/data/itbook-export.pdf)).
>
> - The link between distortion (related to weights) and deviation (pertaining to output) is unclear.
>
> The link between the distortion and the deviation is given in Proposition 1 proof. Specifically, we prove that both the distortion and the deviation scale with the quantization rate as $O(1/k^2)$. To clarify this link, Proposition 1 now explicitly states this connection in the revised version.

---

### Official Review · Reviewer_4eGc · 2023-10-30

**Soundness:** 3 good
**Presentation:** 3 good
**Contribution:** 2 fair
**Rating:** 5
**Confidence:** 5

**Summary:**

This paper proposes a post-training quantization algorithm named Rotation-Invariant Quantization (RIQ) to quantify the NN to mixed-precision, and the main approach is picking the quantization bin width to be proportional to the layers’ norm. Based on the rate-distortion theory, the proposed method searching for the optimal solution over the family of spherical distributions. Empirical results show the competitive performance of RIQ on several benchmarks.

**Strengths:**

1.The authors provide a detailed analysis of the rate-distortion theory, which clarifies their research motivation well.

2.This paper is well-written and organized, and the supplementary material is sufficiently detailed.

**Weaknesses:**

1. Mixed precision is difficult to apply in the industrial scenarios. It usually requires the design of specialized chips to achieve a slight increase in inference speed, so I have doubts about the impact of the proposed method.

2. The experimental result lacks a comparison of inference speed of compressed model, between the proposed method and existing works.

3. The experimental result lacks a comparison of lightweight structure including separable convolution (like mobilenetv2) with existing methods like AdaRound or BRECQ.

**Questions:**

1. I suggest the author provide more descriptions about the scenarios where the mixed-precision model can be applied. The advantages of mixed-precision models can be manifested in NLP-type structures.

---

> ### Author Response · Authors · 2023-11-14
>
> *We thank the reviewer for the time and effort and useful feedback!*
>
> - Mixed precision is difficult to apply in the industrial scenarios. It usually requires the design of specialized chips to achieve a slight increase in inference speed, so I have doubts about the impact of the proposed method
>
> [A] This study does not target any specific hardware and allows running on standard CPU/GPU with 8 bits after de-quantizing the weights. By doing this, we do not benefit from acceleration from the mixed precision (only from the 8-bit execution), and the main benefit is a reduced model size.
>
> One may consider a specialized hardware solution, but in practice, if all symbols are encoded with less than 8 bits, it allows running the model with 8-bit computation, rather than using a mixed-bit on dedicated hardware and achieving a slight increase in inference speed, as the reviewer mentioned. Accordingly, it is not necessary to restore the weights to full precision. Our compression scheme facilitates such quantization, as we point out in the Remark below Proposition 2.
>
> Still, we would like to emphasize that our goal is to reach the smallest model size rather than accelerate the running time, and it should be the main takeaway from this study.
>
> - The experimental result lacks a comparison of inference speed of the compressed model, between the proposed method and existing works.
>
> [A] As mentioned, the objective of this study is to minimize the model size under the cosine distance constraint. In Section A.8.2, we did perform full quantization (i.e., quantizing weights by RIQ and the activations by [Wu 20'](https://arxiv.org/pdf/2004.09602.pdf)), where the de-quantized weights were packed in 8-bits, allowing to further accelerate the inference. Nevertheless, the same speedup was obtained also by a simple uniform 8-bit quantization. So, the main benefit of RIQ is the reduced model size and not its acceleration.
>
> - The experimental result lacks a comparison of lightweight structure including separable convolution (like mobilenetv2) with existing methods like AdaRound or BRECQ.
>
> [A] Please note that Table 1 provides comparisons with the AdaRound and BERCQ on the Resnet50 model.
> RIQ is a powerful method for models with high redundancy and attains good compression results without further tuning. MobileNet_v2 is a relatively compact and efficient model (with a size of only 13MB), and hence, it cannot benefit significantly from RIQ compared to the BRECQ method, which applies further optimization for tuning the weights and requires a few hours to tune on GPU.
>
> As per the reviewer's request, we evaluated RIQ on mobilenet_v2 on ImageNet1k, and the results are given below:
>
> | Method   | compression   | Accuracy(%) | Reference(%) | Drop (%) |
> |----------|---------------|-------------|--------------|----------|
> | BRECQ    | $\times 8$    | $71.66$     | $72.49$      | $0.83$   |
> | AdaRound | $\times 8$    | $69.78$     | $72.49$      | $2.71$   |
> | RIQ      | $\times 5.33$ | $71.19$     | $71.87$      | $0.68$   |
>
>
> - I suggest the author provide more descriptions about the scenarios where the mixed-precision model can be applied. The advantages of mixed-precision models can be manifested in NLP-type structures.
>
> [A] We thank the reviewer for this suggestion. Indeed, NLP-type structures and Large-Language Models (LLMs) in particular, have high redundancies that RIQ can exploit. Indeed, the benefit of a reduced model size is twofold. First, the low memory that is required to store the model. Second, the lower loading and extracting time of the model weights may be significant in LLMs. For example, Llama2-70B requires 140GB to load on the device. Whereas, assuming $\times 4$ compression with RIQ (as demonstrated in Table 4 in the appendix for Llama-7B) would require only 35GB, which can fit a single GPU with 40GB memory (e.g., Nvidia A100).
>
> We added this motivation in the introduction of the revised version.

---

### Official Review · Reviewer_9KPJ · 2023-10-31

**Soundness:** 2 fair
**Presentation:** 2 fair
**Contribution:** 2 fair
**Rating:** 5
**Confidence:** 2

**Summary:**

The authors proposed a post-training mixed-precision weight quantization technique for neural networks by optimization based on an information-theoretic paradigm.  They choose to minimize layer-wise bitwidth constrained by cosine distance.  They experimented with example models in comparison with other methods.

**Strengths:**

- The paradigm of optimization for mixed-precision quantization is novel.
- Theoretical results on optimization bounds are useful.

**Weaknesses:**

- How do layer-wise quantization errors accumilate?  The proposed algorithm does not seem to address this.
- In order for post-training quantization to be practical, activations are quantized too.  How activation quantization can be jointly done is not addressed.
- Experimental results did not show a definitive advantage over competing methods.

**Questions:**

See above.

---

> ### Author Response · Authors · 2023-11-14
>
> *We thank the reviewer for the time and effort and useful comments!*
>
> - How do layer-wise quantization errors accumulate? The proposed algorithm does not seem to address this.
>
> [A] The deviation between the outputs dictates the actual performance, and hence, this is the focus of our study.
>
> In Section 3.1, we do mention that "each quantized layer produces a rotation distortion in its output, and this distortion keeps propagating and accumulating through the layers until reaching the model’s output". In other words, like wave propagation, due to the random nature of errors, the rotations (errors) may accumulate constructively or destructively, and the accumulated errors determine the deviation, and hence, the accuracy.
> Nevertheless, this study does provide a link between the layer-wise errors and the final deviation in Proposition 1. Specifically, this proposition shows that both the final deviation and the layer-wise errors decrease together with the rate as $O(1/k^2)$.
>
> In the revised version, we stress this point out before Proposition 1.
>
> - In order for post-training quantization to be practical, activations are quantized too. How activation quantization can be jointly done is not addressed.
>
> [A] We refer the reviewer to Section A.8.2, where we perform full quantization (i.e., quantizing weights by RIQ and the activations by [Wu 20'](https://arxiv.org/pdf/2004.09602.pdf)). To gain speed on GPU, the de-quantized weights were packed in 8 bits, allowing acceleration of the inference. This way, one can attain a small memory footprint of RIQ and still benefit from full quantization speedup.
>
> - Experimental results did not show a definitive advantage over competing methods.
>
> [A] Table 1 shows that RIQ outperforms many of the methods, achieving a small accuracy drop. Note that there are other methods that also attain small degradation from the baseline, thus, it is hard to show a definitive advantage in this setting.
>
> However, as mentioned in Section A.8.1, RIQ is an efficient method for reducing the model size, and it optimizes models quickly. For example, compressing the ResNet-50 takes less than a minute on a CPU, while most of the methods are more complex and require a longer time to optimize.

---

### Official Review · Reviewer_N6Nd · 2023-11-01

**Soundness:** 3 good
**Presentation:** 3 good
**Contribution:** 3 good
**Rating:** 8
**Confidence:** 5

**Summary:**

The paper proposes a new post-training quantization algorithm that given a neural network and some calibration images, produces mixed-precision quantized network. The key contribution of the paper is a new analysis technique motivated by the cosine-similarity based distortion measure between outputs quantized and unquantized network. The authors provide rate distortion analysis under the proposed measure and find the relation between the quantization bin width ($delta$) and layer's distortion (lemma 1) and the whole model's distortion (corollary 1). The authors then re-parametrize the search over $delta$ as search over parameter $k$ defined wrt distortion, and provide an efficient search algorithm (alg.1). Authors bound the possible values for optimal $k$ and make a heuristic search. The effectiveness of search algorithm, and of the proposed analysis is demonstrated on multiple networks and datasets.

**Strengths:**

- The technique is well motivated and executed; even without the application to compression, the results and its analysis have their merits on its own.
- The heuristic search over $k$ (alg 1), has good initial parameters and does not seem to require heavy tuning.
- Very good presentation and flow.

**Weaknesses:**

- I found it a bit hard to understand how $k$ come into picture during the first readthrough of the paper. I feel like a little more work needed to introduce it.
- There are a few arguable points that I count as a weakness, but they are easily addressable:
  - I would like to understand how good is the search parameters wrt a synthetically created problem where (say weights sampled from Gaussian/Laplacian), single, layer and etc. How much the heuristics during the search may (or may not) miss the optimal bin width?
  - Also, while I agree that in general setting search of $k$ is unbounded (as you write in the paper), practically speaking it is not the case: the weights are finite, and thus there are only certain number of  $delta$-s to check. This, has in fact been done in the work called "Optimal quantization using scaled codebook" (btw, you cite this paper but attribute it as QAT, which is not correct)
- Results:
  - I believe all compression results are given after ANS encoding; providing the compression ratio before ANS would be of great value (most papers report results before any additional encodings)

**Questions:**

Please see weaknesses section.

---

> ### Author Response · Authors · 2023-11-16
>
> *Thank you for the supportive comments and insightful feedback!*
>
> - I found it a bit hard to understand how $k$ came into the picture during the first readthrough of the paper. I feel like a little more work is needed to introduce it.
>
> [A] We agree with the reviewer and the following text was added to introduce $k$:
>
> > The connection of $\Delta_\ell$ to $\Vert w_\ell \Vert$ in Lemma1 hints at the rotation-invariant nature of the optimization. To focus on rotation-invariant solutions, RIQ introduces a search parameter $k$ that maintains proportion with $\Vert w_\ell \Vert$, allowing efficient search over these solutions.  Specifically, when $\Delta_\ell(k) = \Vert w_\ell \Vert/k$  where $k$ to be optimized, the bin-width...
>
> - There are a few arguable points that I count as a weakness, but they are easily addressable:
>     - I would like to understand how good is the search parameters wrt a synthetically created problem where (say weights sampled from Gaussian/Laplacian), single layer, etc. How much the heuristics during the search may (or may not) miss the optimal bin width?
>
>     - Also, while I agree that in the general setting search of $k$ is unbounded (as you write in the paper), practically speaking it is not the case: the weights are finite, and thus there are only a certain number of $\Delta$-s to check. This has in fact been done in the work called "Optimal quantization using scaled codebook" (btw, you cite this paper but attribute it as QAT, which is not correct)
>
> [A1] RIQ performs constrained optimization of parameter $k$, where the constraint is the deviation between outputs. The heuristic allows tolerance from the optimal deviation which is specified by the smallest step.
>
> As suggested by the reviewer, we tested a synthetic problem, where RIQ quantizes a single convolution layer with a Gaussian input tensor to examine the searching points and the possible miss (See results in [this Figure](https://anonymous.4open.science/r/riq-3CE1/conv2d_deviation.pdf)). In this example, the deviation constraint between the outputs is 0.005 and RIQ reached very near with a deviation value of 0.00499 (i.e., missed by 1e-5).
>
> [A2]  Indeed, practically there's a bounded search space in which the optimization can be done.  In the revised text we rephrase the text below eq. (7):
>
> > Even though the search of $k$ is unbounded in general, practically it is sufficient to search in bounded space since the weights' norm is finite (Idelbayev et al. 2021).
>
> Thank you for pointing out the incorrect citation of this work. We fixed it in the revised version.
>
> - I believe all compression results are given after ANS encoding; providing the compression ratio before ANS would be of great value (most papers report results before any additional encodings)
>
> [A] Indeed, the results are after applying ANS encoding, and we agree that there is great value in seeing the contribution of ANS to RIQ alone, and hence, we provide the rate before and after ANS for the ResNet50 model in Figure 4(a), from which we can infer the contributions of ANS to Table 1.
>
> Figure 4(a) shows that the entropy limit is 2-3 bits/symbol away from RIQ only (depends on the deviation requirement). This difference stems from the different representations' efficiency. In other words, the RIQ representation minimizes the entropy without attaining it, whereas the ANS encoding attains the entropy limit.
>
> Please note that while other baselines did not apply further encoding, most of the baselines have used per-channel quantization, for which encoding may be less beneficial due to the encoding tables overhead.

---

### Author Response · Authors · 2023-11-21

First, we would like to emphasize that we highly appreciate the hard work done and fully respect the professional opinion of the reviewers. We want to thank all the reviewers for their time and constructive and valuable feedback. Both the structure and the presentation of this paper have been improved based on the thorough reviews and insightful comments made by the reviewers. We enclose a revision of the paper, which we believe addresses all the comments.

The revised version has a few fundamental differences compared to the previous one (all revisions are marked in blue color for tractability). We first outline the main updates. We then specifically address questions raised by the reviewers and detail the revisions made.

We improved the structure and presentation of the paper based on the reviewers' suggestions. In particular:

 * We rephrased Lemma 1, explicitly mentioning that it holds for the high-rate regime. The proof is revised accordingly.

 * We stressed the motivation for using RIQ in large language models, where redundancy is relatively high and memory is a bottleneck.

 * We emphasize the link between the layer-wise errors and the final deviation given in Corollary 1 and Proposition 1.

 * All revisions are marked in blue color for tractability

---

### Meta-Review · Area_Chair_Y4dh · 2023-12-09

**Metareview:**

The submission proposes a rotation invariance perspective on network quantization for compression.  The submission received mixed reviews with one enthusiastic review, and three reject recommendations.  The highest and lowest scores were also the most substantive reviews, though there were valid points raised by the other reviewers about layerwise accumulation of errors and the questionability of measuring mixed precision size without measuring latency on actual hardware.  On the balance, a majority of reviewers had valid concerns.

**Justification For Why Not Higher Score:**

Majority of reviewers felt the paper was not ready for publication.  Open questions to make theory precise, and to evaluate latency.

**Justification For Why Not Lower Score:**

N/A

---

### Decision · Program_Chairs · 2024-01-16

Reject